# η-pairing state in a flat band lattice: interband coupling effects on entanglement entropy logarithm

**Seik Pak[1], Hanbyul Kim[1], Chan Bin Bark[1], Sang-Jin Sin[1][*], Jae-yoon Choi[2][†] and Moon Jip Park[1,3][‡]**

**1** Department of Physics, Hanyang University, Seoul 04763, Republic of Korea
**2** Department of Physics, Korea Advanced Institute of Science and Technology, Daejeon 34141, Korea
**3** Research Institute for Natural Science and High Pressure, Hanyang University, Seoul 04763, Republic of Korea

[*] sjsin@hanyang.ac.kr,   [†] jaeyoon.choi@kaist.ac.kr,   [‡] moonjippark@hanyang.ac.kr

## Abstract

The $\eta$-pairing states are the exact eigenstates of the hypercubic Hubbard model, which exhibit an anomalous logarithmic scaling of entanglement entropy. **It has been shown that $\eta$-pairing states are still exact eigenstates even in multi-band systems, provided interband coupling is negligible on the flat band [1,2].** However, typical flat band systems such as Lieb and Kagome lattices often feature band touchings, where interband coupling effects play a role. As an example of such a system, we investigate the deformation of $\eta$-pairing states in the Creutz ladder model with the interband coupling effect. **Our results show that the interband coupling introduces effective non-local interactions within the flat band. In contrast to previous findings in isolated flat band systems.** We observe the suppression of entanglement entropy and the violation of the spectrum generating algebra. Interestingly, these effects persist even in the limit of infinitesimally small interband coupling. These findings illuminate the robustness and limitations of $\eta$-pairing in realistic setups, offering insights into the physics of interacting flat band systems.

## Contents

---

# 1 Introduction

Entanglement entropy (EE) is a fundamental diagnostic of quantum correlations in many-body quantum states [3–10]. Typically, the EE of thermal states obeys a volume law, scaling proportionally to the system size ($\sim L^d$), while the ground states of gapped Hamiltonians follow an area law ($\sim L^{d-1}$). On the other hand, certain critical systems [11–16] exhibit a logarithmic scaling of EE ($\sim \log L$), reflecting the presence of long-range correlations and emergent symmetry structures, such as quantum many-body scars and Hilbert space fragmentation [6, 14, 17–23].

A representative example of such emergent structures is the $\eta$-pairing state, first introduced by C. N. Yang in the context of the single-band Hubbard model [24]. A hidden pseudo-spin SU(2) symmetry enforces perfectly bound Cooper pairs as eigenstates, giving rise to the off-diagonal long-range order (ODLRO) of a doublon pairing order. The resulting $\eta$-paired eigenstates form an equally spaced tower of eigenenergies. In multi-band systems, recent work has shown that $\eta$-pairing can also be stabilized in flat band systems, provided the flat band is perfectly isolated from other bands [1, 2].

However, in many lattice realizations of flat bands such as the Lieb, Kagome, and Creutz ladder models [25], the flat band is accompanied by symmetry-protected band touching with dispersive bands [26, 27]. This band touching induces non-negligible couplings between the flat and dispersive bands, thereby breaking the exact pseudo-spin SU(2) symmetry that underpins the ideal mechanism of $\eta$-pairing states. Consequently, it remains an open question how these states and their associated phenomena, such as energy spacing, correlation length, and EE, evolve under interband interactions.

In this work, we address the modification of ideal $\eta$-pairing states in the Creutz ladder, a representative flat band model in one dimension. Our results diverge from previous studies [1] by explicitly investigating the effects of the dispersive band. We reveal three key phenomena. First, while the interband coupling disrupts the exact pseudo-spin SU(2) symmetry and the associated spectrum generating algebra (SGA) of the $\eta$-pairing states, we capture the modified energy shifts induced by virtual excitations through interband coupling. Second, in the real-

space picture, the size of the doublon in the perturbed states increases, characterized by the delocalization of the pair correlation function. Third, we demonstrate that interband coupling introduces mobility to the electrons on the flat band and show that the dominant interaction between doublons is repulsive.

Recent studies have further revealed that $\eta$-pairing states are intimately connected to quantum many-body scars (QMBS), a class of nonthermal eigenstates that coexist with a predominantly thermal spectrum [10, 28–34]. Importantly, interband coupling can bridge these otherwise isolated sectors, effectively modifying the interactions between macroscopic degrees of freedom and, consequently, the overall thermalization dynamics. Finally, we discuss experimental realizations of our results in cold-atom systems.

## 2  Model and Theoretical Setup

### 2.1  $\eta$-pairing states in hypercubic lattice

We start our discussion by introducing the $\eta$-pairing state in the $D$-dimensional hypercubic lattice Hubbard model. The Hamiltonian is given by

$$H_{\text{cubic}} = t \sum_{\langle \mathbf{r},\mathbf{r}'\rangle,\sigma} c_{\mathbf{r}\sigma}^\dagger c_{\mathbf{r}'\sigma} + U \sum_{\mathbf{r}} n_{\mathbf{r}\uparrow} n_{\mathbf{r}\downarrow} - \mu \sum_{\mathbf{r},\sigma} c_{\mathbf{r}\sigma}^\dagger c_{\mathbf{r}\sigma},$$

where $c_{\mathbf{r}\sigma}$ ($c_{\mathbf{r}\sigma}^\dagger$) denotes the fermionic annihilation (creation) operator at site $\mathbf{r}$ with spin $\sigma$, and $n_{\mathbf{r}\sigma} = c_{\mathbf{r}\sigma}^\dagger c_{\mathbf{r}\sigma}$ represents the number operator. Here, $\langle \mathbf{r},\mathbf{r}'\rangle$ denotes nearest-neighbor pairs.

Assuming periodic boundary conditions (PBC), we can define the $\eta$-operator, which forms a doublon pair with center-of-mass momentum $\boldsymbol{\pi} = (\pi, \pi, \ldots, \pi)$, as:

$$\eta = \sum_{\mathbf{r}} e^{i\boldsymbol{\pi}\cdot\mathbf{r}} c_{\mathbf{r}\uparrow} c_{\mathbf{r}\downarrow}. \tag{1}$$

The Hubbard model preserves pseudo-spin SU(2) symmetry. The generators of the pseudo-spin SU(2) symmetry, defined as $\eta_x = \frac{1}{2}(\eta + \eta^\dagger)$, $\eta_y = \frac{i}{2}(\eta - \eta^\dagger)$, and $\eta_z = \frac{1}{2}[\eta^\dagger, \eta]$, satisfy the following symmetry algebra:

$$[\eta_a, \eta_b] = i\epsilon_{abc}\eta_c, \quad [H, J^2] = 0, \quad [H, \eta_z] = 0, \tag{2}$$

where $J^2 = \eta_x^2 + \eta_y^2 + \eta_z^2$ and indices $a, b, c$ range over $x, y, z$. Accordingly, the $\eta$ operators satisfy the following commutation relation (also known as the SGA [28]):

$$[H, \eta^\dagger] = (U - 2\mu)\eta^\dagger. \tag{3}$$

Due to the SGA, the eigenenergies of the Hubbard model form an equally spaced tower of eigenstates:

$$|\psi_n\rangle = \left(\eta^\dagger\right)^n |\psi_0\rangle, \tag{4}$$

where $|\psi_0\rangle$ is the vacuum (or a reference eigenstate) and $n$ runs over the possible number of pairs. Intuitively, each application of $\eta^\dagger$ creates a doublon pair, raising the energy by $U - 2\mu$ due to the on-site interaction and chemical potential. This original formulation of $\eta$-pairing, characterized by a macroscopic doublon condensate with ODLRO, sets the stage for our investigation.

The integrability of $\eta$-pairing results in logarithmic EE behavior. This feature can also be seen from the well-defined quasi-particle behavior of $\eta$-pairing states appearing in Eq. 4 [17].

From the homogeneous spatial configuration of the doublons, one can write the reduced density matrix of the $\eta$-pairing states as follows:

$$\rho_A = \sum_{i=0}^{L_A} \lambda_i \, |i\rangle\langle i|, \quad \lambda_i = \frac{\binom{L_A}{i}\binom{L_B}{n-i}}{\binom{L}{n}}, \tag{5}$$

where $L = L_A + L_B$ is the total system size, $L_A$ and $L_B$ denote the sizes of subsystems $A$ and $B$, respectively, and $|i\rangle$ represents a symmetric state with $i$ doublons in $L_A$ sites [35, 36]. The EE is then given by $S = -\sum_{i=0}^{L_A} \lambda_i \ln \lambda_i$. In the thermodynamic limit, this sum can be evaluated analytically, yielding

$$S = \frac{1}{2}\Big(1 + \ln\big[2\pi \tfrac{n}{L}\big(1 - \tfrac{n}{L}\big)L_A\big]\Big), \tag{6}$$

which shows a characteristic logarithmic dependence on $L_A$ [37].

## 2.2 Multiband flat band systems

To capture the essential physics of $\eta$-pairing in flat band systems, we consider a Hamiltonian of the form $H_{\text{total}} = H_{\text{kin}} - \mu N + H_{\text{int}}$, where $H_{\text{kin}}$ denotes the single-particle tight-binding model and $H_{\text{int}}$ the on-site interaction term. Without loss of generality, the tight-binding model can be written in momentum space as:

$$H_{\text{kin}} = \sum_{n=1, s=\uparrow,\downarrow}^{N_{\text{orb}}} \sum_{\mathbf{k}\in\text{BZ}} \epsilon_n(\mathbf{k}) \gamma_{n\mathbf{k}s}^\dagger \gamma_{n\mathbf{k}s}, \tag{7}$$

where $\epsilon_{n,s}(\mathbf{k})$ is the eigenenergy of the $n$-th band with momentum $\mathbf{k}$ and spin $s = \uparrow, \downarrow$. $\gamma_{n\mathbf{k}s}$ is the annihilation operator of the $n$-th single-particle eigenstate, related to the electron operator by $c_{\alpha\mathbf{k}s} = [U(\mathbf{k})]_{\alpha,n}\gamma_{n\mathbf{k}s}$, with the orbital index $\alpha$.

Without loss of generality, we assume that $n = 1$ corresponds to the flat band while the other bands are dispersive. We consider the projection of the full Hilbert space onto the flat band subspace. The projection operator is defined as $P_{\alpha\beta}(\mathbf{k}) = [U(\mathbf{k})]_{\alpha,1}[U(\mathbf{k})^\dagger]_{1,\beta}$, and the Fourier transform yields the corresponding projected electron operator in real space:

$$\begin{aligned}
\bar{c}_{i\alpha\sigma} &= \frac{1}{\sqrt{N}} \sum_{\mathbf{k},\beta} e^{-i\mathbf{k}\cdot\mathbf{R}_i} P_{\alpha\beta}(\mathbf{k}) c_{\mathbf{k}\beta\sigma} \\
&= \sum_{j,\beta} P_{\alpha\beta}(i-j) c_{j\beta\sigma}, \tag{8} \\
\tilde{c}_{i\alpha\sigma} &= c_{i\alpha\sigma} - \bar{c}_{i\alpha\sigma}, \tag{9}
\end{aligned}$$

where $\bar{c}_{i\alpha\sigma}$ and $\tilde{c}_{i\alpha\sigma}$ denote the projected annihilation operators in the flat band and the complementary Hilbert-Fock space, respectively. It is important to note that the projected operators satisfy the following anticommutation relations:

$$\begin{aligned}
\{\bar{c}_{i\alpha\sigma}, \bar{c}_{j\beta\sigma'}^\dagger\} &= \delta_{\sigma\sigma'} P_{\alpha\beta}(i-j), \\
\{\tilde{c}_{i\alpha\sigma}, \tilde{c}_{j\beta\sigma'}^\dagger\} &= \delta_{\sigma\sigma'}\big(\delta_{\alpha\beta} - P_{\alpha\beta}(i-j)\big), \\
\{\bar{c}_{i\alpha\sigma}, \tilde{c}_{j\beta\sigma'}^\dagger\} &= 0. \tag{10}
\end{aligned}$$

We can define the $\bar{\eta}$-operator with the projected fermion operators in the flat band subspace as:

$$\bar{\eta} = \sum_{i,\alpha} \bar{c}_{i\alpha\uparrow}\bar{c}_{i\alpha\downarrow}. \tag{11}$$

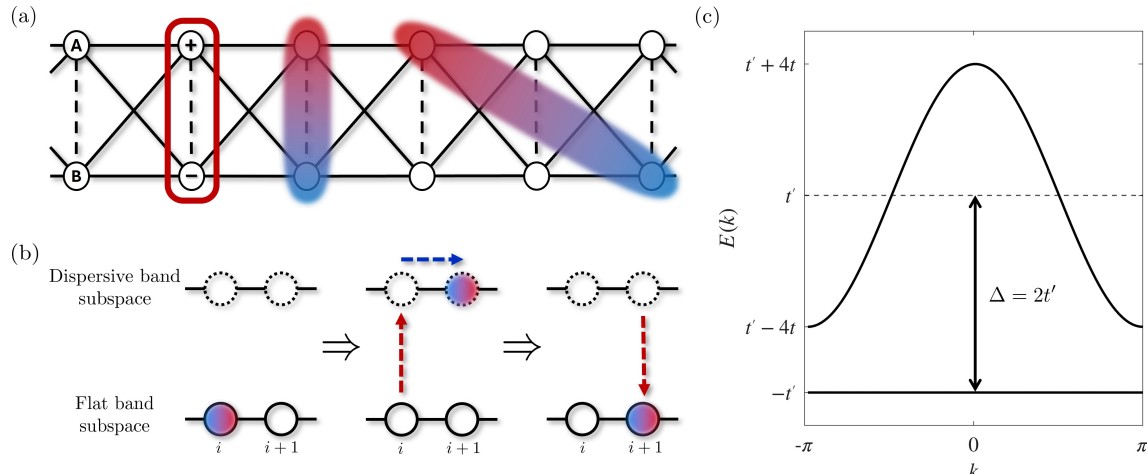

Figure 1: (a) Lattice structure of the Creutz ladder, consisting of two sublattices $A$ and $B$. Solid lines represent intercell hopping and dashed lines denote intracell hopping of amplitudes $t$ and $t'$, respectively. The red box highlights the CLS of the flat band. Shaded regions represent the spatial profile of up-spin and down-spin pair in the doublon. The vertical shaded region illustrates the localized doublon characteristic of the $\bar{\eta}$-pairing state, while the tilted shaded region shows the spatially extended doublon associated with the $\eta^*$-pairing state arising from interband coupling. (b) Schematic representation of the effective doublon hopping process from site $i$ to the neighboring site $i+1$. Solid circles represent the effective real-space sites of the flat band subspace, while dashed circles denote the corresponding sites in the dispersive band subspace. Red arrows indicate virtual interband excitation and relaxation processes (creating or annihilating virtual doublons), while the blue dashed arrow represents the hopping process of the doublon within the dispersive band. (c) Band structure of the Creutz ladder when $t = 1$ and $t' = 3$ with dispersion relations $E_{\text{disp}}(k) = t' + 4t \cos k$, $E_{\text{flat}} = -t'$, and band gap $\Delta = 2t'$.

However, in generic multi-band systems, the SU(2) symmetry is not justified. We can explicitly see this from the modification of the commutation relation as,

$$
\begin{aligned}
[H_{\text{total}}, \bar{\eta}^\dagger] =& (U - 2\mu)\bar{\eta}^\dagger - U\Big[\sum_{j,\beta} \sum_{\substack{m \neq n \\ \gamma \neq \delta}} P_{\gamma\beta}(j-m) c^\dagger_{m\gamma\uparrow} P_{\delta\beta}(j-n) c^\dagger_{n\delta\downarrow} \\
& + \sum_{ij\alpha,\beta} P_{\alpha\beta}(i-j)\big(n_{i\alpha\uparrow} \bar{c}^\dagger_{j\beta\uparrow} c^\dagger_{i\alpha\downarrow} + n_{i\alpha\downarrow} c^\dagger_{i\alpha\uparrow} \bar{c}^\dagger_{j\beta\downarrow}\big)\Big].
\end{aligned}
\tag{12}
$$

The second term in Eq. (12) is the source of non-integrability. It has been shown that one can define a generalized $\eta$-pairing in certain types of lattice systems, including bipartite systems [28]. However, as we demonstrate below, general flat band systems do not fall into this category.

## 2.3 Schrieffer-Wolff transformation of $\eta$-pairing

As a concrete example, we consider the one-dimensional Creutz ladder model that hosts a flat band (See Fig. 1(a)) with the onsite Hubbard interaction:

$$H_{\text{kin}} = \sum_{i,\sigma} \Big[ t \big( c_{i+1,A,\sigma}^{\dagger} c_{i,A,\sigma} + c_{i+1,B,\sigma}^{\dagger} c_{i,A,\sigma} + c_{i+1,A,\sigma}^{\dagger} c_{i,B,\sigma} + c_{i+1,B,\sigma}^{\dagger} c_{i,B,\sigma} \big) + t' c_{i,B,\sigma}^{\dagger} c_{i,A,\sigma} + \text{h.c.} \Big],$$

$$(13)$$

$$H_{\text{int}} = U \sum_{i,\alpha=A,B} n_{i\alpha\uparrow} n_{i\alpha,\downarrow},$$

$$(14)$$

where $\alpha$ and $i$ denote the sublattice and site index, respectively. The kinetic part of the Hamiltonian consists of intra-cell hopping $t'$ between sublattices A and B within the same unit cell, and inter-cell hopping $t$ between nearest unit cells. Without the Hubbard interaction, the Hamiltonian describes a two-band system with dispersion relations $E_{\text{disp}}(k) = t' + 4t \cos k$, $E_{\text{flat}} = -t'$, and band gap $\Delta = 2t'$ (Fig. 1(c)). The eigenstate of the flat band is given as $U(\mathbf{k}) = (1, -1)^{\text{T}}/\sqrt{2}$. The momentum independence of the flat band eigenstate renders the projection operator $P_{\alpha\beta}$ momentum independent. Consequently, the local projected fermion annihilation operator can be written as:

$$\bar{c}_{i\alpha\sigma} = \frac{1}{2} \big( c_{i\alpha\sigma} - c_{i\alpha'\sigma} \big),$$

$$(15)$$

which simplifies to a local linear combination of the original operators, where $\alpha'$ denotes the opposite sublattice index of $\alpha$. In other words, $\bar{c}_{i\alpha\sigma}$ corresponds to the compact localized state (CLS) of the Creutz ladder model.

We derive the effective Hamiltonian in the flat band subspace using the Schrieffer–Wolff (SW) transformation by treating the effect of $H_{\text{int}}$ perturbatively [38]. Here, we define the many-body projection operator $\hat{\mathcal{P}}$ onto the flat band space and its orthogonal projector $\hat{\mathcal{Q}} = \mathbf{1} - \hat{\mathcal{P}}$ onto the complementary space, such that the Fock state $|\psi\rangle = \gamma_{n_1 \mathbf{k}_1 s_1}^{\dagger} \cdots \gamma_{n_{n_f} \mathbf{k}_{n_f} s_{n_f}}^{\dagger} |0\rangle$ satisfies $\hat{\mathcal{P}}|\psi\rangle = |\psi\rangle$ if all $n_i = 1$, and zero otherwise. The effective Hamiltonian can be perturbatively expanded as:

$$H_{\text{eff}} = (H_{\text{kin}} - \mu N)\hat{\mathcal{P}} + \hat{\mathcal{P}} H_{\text{int}} \hat{\mathcal{P}} + \frac{1}{2} \hat{\mathcal{P}} [\mathcal{L}(H_{\text{int}}), \mathcal{O}(H_{\text{int}})] \hat{\mathcal{P}} + \cdots,$$

$$(16)$$

where the superoperators $\mathcal{O}(X)$ and $\mathcal{L}(X)$ acting on an operator $X$ are defined as:

$$\mathcal{O}(X) = \hat{\mathcal{P}} X \hat{\mathcal{Q}} + \hat{\mathcal{Q}} X \hat{\mathcal{P}}, \quad \mathcal{L}(X) = \sum_{i,j} \frac{|i\rangle \langle i| \mathcal{O}(X) |j\rangle \langle j|}{E_i - E_j},$$

where $|i\rangle$ and $|j\rangle$ are eigenstates of the unperturbed Hamiltonian $H_{\text{kin}}$, and $E_i, E_j$ are the corresponding eigenenergies. The operator $\mathcal{O}(X)$ extracts the off-diagonal components of $X$ that connect states in the flat band and complementary subspaces, ensuring that the indices $i$ and $j$ in the summation belong to different subspaces. This guarantees that the energy denominator $E_i - E_j$ is always nonzero, as long as the energy gap between the flat band and the dispersive band exists. Physically, $\mathcal{L}(X)$ incorporates the effects of virtual hopping processes between the flat band and the complementary bands, with each contribution weighted by the inverse energy difference, thereby capturing higher-order corrections to the effective low-energy dynamics.

For a perfectly isolated flat band (i.e., infinite energy gap between the flat band and dispersive band), only the first two terms on the right-hand side of Eq. (16) are non-vanishing:

$$
\begin{aligned}
H_{\text{eff}}^{(1)} &\approx (H_{\text{kin}} - \mu N)\hat{\mathcal{P}} + \hat{\mathcal{P}} H_{\text{int}} \hat{\mathcal{P}} \\
&= (H_{\text{kin}} - \mu N)\hat{\mathcal{P}} + U \sum_{i\alpha} \bar{c}_{i\alpha\uparrow}^{\dagger} \bar{c}_{i\alpha\downarrow}^{\dagger} \bar{c}_{i\alpha\downarrow} \bar{c}_{i\alpha\uparrow} \hat{\mathcal{P}} \\
&= (H_{\text{kin}} - \mu N)\hat{\mathcal{P}} + U \sum_{i\alpha} \bar{n}_{i\alpha,\downarrow} \bar{n}_{i\alpha\uparrow} \hat{\mathcal{P}},
\end{aligned}
$$

$$(17)$$

where $\bar{n}_{i\alpha\sigma} = \bar{c}^\dagger_{i\alpha\sigma} \bar{c}_{i\alpha\sigma}$. Since $(H_{\text{kin}} - \mu N)\hat{\mathcal{P}}$ is a constant times the number operator, we can redefine the term with an effective chemical potential as $(H_{\text{kin}} - \mu N)\hat{\mathcal{P}} = \mu_{\text{eff}} N$. Then, from Eq. (10), one can verify that $H^{(1)}_{\text{eff}}$ preserves SU(2) symmetry, and the projected $\eta$ operator satisfies the SGA with the modification as:

$$[H^{(1)}_{\text{eff}}, \bar{\eta}^\dagger] = \left( \frac{U}{2} - 2\mu_{\text{eff}} \right) \bar{\eta}^\dagger, \tag{18}$$

with

$$\bar{\eta}^\dagger = \sum_{i,\alpha} \bar{c}^\dagger_{i\alpha\uparrow} \bar{c}^\dagger_{i\alpha\downarrow}. \tag{19}$$

As in Sec. 2.1, $\bar{\eta}^\dagger$ generates towers of eigenstates; among them, we define the $\bar{\eta}$-pairing state as the tower of eigenstates starting from the vacuum, namely:

$$|\bar{\eta}_n\rangle = (\bar{\eta}^\dagger)^n |0\rangle. \tag{20}$$

In contrast, when the flat band is not perfectly isolated, second-order and higher-order terms from the SW transformation become significant. These corrections modify the effective Hamiltonian and disrupt the exact commutation relations of the SGA as shown in Eq. (12), such that $\bar{\eta}^\dagger$ no longer generates exact eigenstates. To compare the actual eigenstate with the $\bar{\eta}$-pairing ansatz, we identify a distinctive subset of eigenstates of the full Hamiltonian, denoted as $|\eta^*_n\rangle$, which retains the essential characteristics of the original $\bar{\eta}$-pairing states. We define these $\eta^*$-pairing states by maximizing the fidelity with their respective $\bar{\eta}$-pairing states $|\bar{\eta}_n\rangle$. Physically, the $\eta^*$-pairing states represent the adiabatic evolution of the $\bar{\eta}$-pairing states as the system is driven from the SU(2)-symmetric regime to the symmetry-broken regime. Indeed, in the limit of $t \to 0$ and $t' \to \infty$, the $\eta^*$-states converge exactly to the $\bar{\eta}$-pairing states: $\langle \bar{\eta}_n | \eta^*_n \rangle \to 1$.

## 2.4 Projected Kinetic Hamiltonian

[Note: This subsection is newly added.]

To analyze the effect of interband coupling beyond the isolated limit, we must account for the effect of the dispersive band within the complementary subspace. We decompose the complementary space projected kinetic Hamiltonian $\hat{\mathcal{Q}} H_{\text{kin}} \hat{\mathcal{Q}}$ into a constant energy gap and a kinetic dispersion term. Defining the energy relative to the flat band energy $E_{\text{flat}}$, we write:

$$\hat{\mathcal{Q}}(H_{\text{kin}} - N E_{\text{flat}})\hat{\mathcal{Q}} = \Delta \sum_{i\alpha\sigma} \tilde{n}_{i\alpha\sigma} + t \sum_{\langle i,j \rangle \sigma} \sum_{\alpha,\beta} \left( \tilde{c}^\dagger_{i\alpha\sigma} \tilde{c}_{j\beta\sigma} + \text{H.c.} \right)$$
$$= \tilde{\Delta}_{\tilde{N}} + \delta\hat{h}, \tag{21}$$

where $N$ is the total number of electrons, $\tilde{n}_{i\alpha\sigma} = \tilde{c}^\dagger_{i\alpha\sigma} \tilde{c}_{i\alpha\sigma}$ is the complementary space number operator, and $\tilde{N} = \sum_{i\alpha\sigma} \tilde{n}_{i\alpha\sigma}$ represents the total number of electrons in the complementary subspace (i.e., the number of virtual particles participating in the perturbation process). $\tilde{\Delta}_{\tilde{N}} = \Delta \tilde{N}$ is the energy difference between the flat band subspace eigenstate and virtually excited states in the dispersive band subspace, while $\delta\hat{h}$ captures the effect of the $k$-dependence of the dispersive band. A detailed derivation is provided in Appendix B.

# 3 Modification of spectrum generating algebra

[Note: This section has been completely rewritten.]

To quantify the deviation from the exact SU(2) symmetry, we employ the SW transformation in Eq. (16). We explicitly treat the propagation of virtual excitations using the resolvent operator formalism (detailed derivations in Appendix B). With the resolvent operator, the second-order effective Hamiltonian can be written as:

$$
H_{\text{SW}}^{(2)} = \frac{1}{2}\hat{\mathcal{P}}[\mathcal{L}(H_{\text{int}}), \mathcal{O}(H_{\text{int}})]\hat{\mathcal{P}} = \hat{\mathcal{P}}H_{\text{int}}\hat{\mathcal{R}}H_{\text{int}}\hat{\mathcal{P}}, \tag{22}
$$

where $\hat{\mathcal{R}}$ is the resolvent operator acting on the complementary subspace $\hat{\mathcal{Q}}$. It is defined as:

$$
\hat{\mathcal{R}} = \hat{\mathcal{Q}}\frac{1}{NE_{\text{flat}} - H_{\text{kin}}}\hat{\mathcal{Q}}. \tag{23}
$$

The resolvent describes the propagation of virtual excitations within the dispersive bands. From Eq. (21), we can expand the resolvent operator in a geometric series with respect to the dimensionless parameter $\delta\hat{h}/\Delta$ (of order $t/t'$) as follows:

$$
\hat{\mathcal{R}} \approx -\frac{1}{\tilde{\Delta}_{\tilde{N}}}\hat{\mathcal{Q}} + \frac{1}{\tilde{\Delta}_{\tilde{N}}}\delta\hat{h}\frac{1}{\tilde{\Delta}_{\tilde{N}}} - \mathcal{O}\left(\frac{t^2}{\tilde{\Delta}_{\tilde{N}}^3}\right). \tag{24}
$$

Substituting this expansion back into Eq. (22), we obtain the effective Hamiltonian decomposed into local and non-local contributions: $H_{\text{SW}}^{(2)} \approx H_{\text{local}}^{(2)} + H_{\text{non}-\text{local}}^{(2)}$. The zeroth order of the geometric series in Eq. (24) gives the local renormalization of the on-site interaction:

$$
H_{\text{local}}^{(2)} = -\frac{U^2}{2\Delta}\sum_{i,\alpha}\bar{n}_{i\alpha\downarrow}\bar{n}_{i\alpha\uparrow}. \tag{25}
$$

The above term effectively renormalizes the Hubbard interaction to $U_{\text{eff}} = U - U^2/(2\Delta)$. Importantly, $H_{\text{local}}^{(2)}$ preserves the SGA, merely modifying the energy spacing of the $\bar{\eta}$-pairing tower:

$$
[H_{\text{eff}}^{(2)}, \bar{\eta}^\dagger] \approx \left(\frac{U_{\text{eff}}}{2} - 2\mu_{\text{eff}}\right)\bar{\eta}^\dagger = \epsilon\bar{\eta}^\dagger. \tag{26}
$$

On the other hand, the second-order term of the geometric series (which scales as $t^2/\Delta^3$) breaks SU(2) symmetry, generating the following effective pair hopping processes described in Fig. 1(b):

$$
H_{\text{non}-\text{local}}^{(2)} \approx -\hat{\mathcal{P}}H_{\text{int}}\hat{\mathcal{Q}}\left(\frac{(\delta\hat{h})^2}{\tilde{\Delta}_2^3}\right)\hat{\mathcal{Q}}H_{\text{int}}\hat{\mathcal{P}} = -J_{\text{eff}}\sum_{\langle i,j\rangle}\eta_i^\dagger\eta_j, \quad \text{with } J_{\text{eff}} = \frac{U^2t^2}{8\Delta^3}, \tag{27}
$$

where $\bar{\eta}_i^\dagger = \sum_\alpha \bar{c}_{i\alpha\uparrow}^\dagger\bar{c}_{i\alpha\downarrow}^\dagger$. This induced kinetic energy $J_{\text{eff}}$ disrupts the SGA by generating dispersion to the doublons.

As we will see in the next section, despite the violation of exact SU(2) symmetry, the modified SGA in the perturbative regimes still supports logarithmic EE scaling. As depicted in Fig. 2(a), as $U/\Delta$ increases and higher-order local corrections $H_{\text{local}}^{(n)}$ become significant, the energy shift deviates from the calculated $\epsilon = \frac{U}{2} - \frac{U^2}{4\Delta}$. Similarly, in Fig. 2(b), the deviation becomes larger than in Fig. 2(a) due to the pair hopping process $H_{\text{non}-\text{local}}^{(2)}$ which breaks SGA. Nonetheless, for $|U| \ll \Delta$, the structure of SGA remains robust enough to produce towers of nearly equal level spacing. As the band gap $\Delta$ decreases (or as $U$ grows larger), the higher-order corrections in SW transformation become non-negligible, resulting in a modification of the overall thermalization behavior.

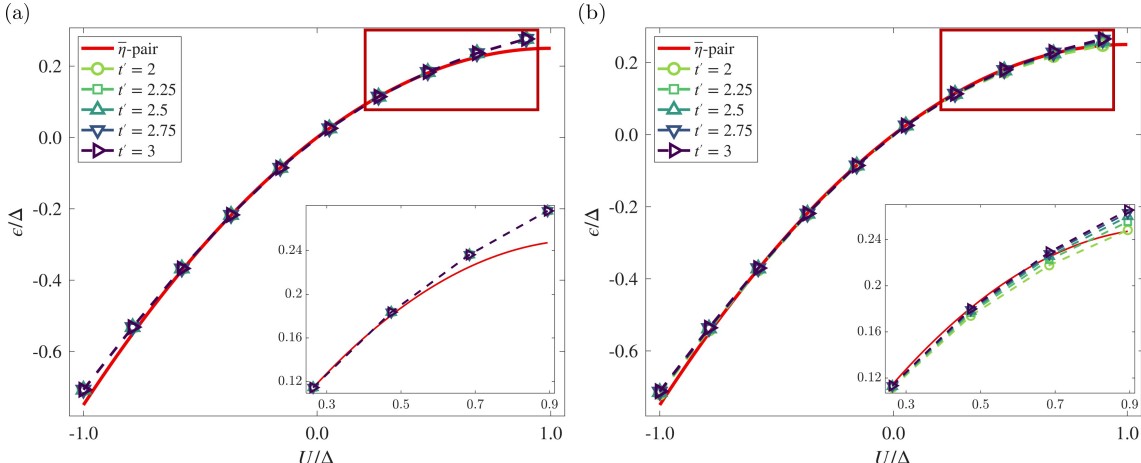

Figure 2: (a) Energy shift of a single $\eta^*$-pair ($|\eta_1^*\rangle$) from the vacuum $\epsilon/\Delta$ as a function of the scaled interaction strength $U/\Delta$ for various values of $t'$ when $t = 0$. In this limit ($\delta\hat{h} \to 0$), numerical simulations (symbols with dashed lines) agree closely with the analytical prediction $\epsilon = \frac{U}{2} - \frac{U^2}{4\Delta}$ (solid line) derived from the local term $H_{\text{local}}^{(2)}$, indicating the validity of the SGA. The inset corresponds to the magnified region highlighted by the red box. (b) Energy shift from the vacuum $\epsilon/\Delta$ for $t = 0.5$. Deviations from the analytical prediction increase as $\delta\hat{h}$ becomes significant (smaller $t'$), highlighting the breakdown of the SGA due to the non-local corrections $H_{\text{non-local}}^{(2)}$.

## 4  Entanglement entropy logarithm

Our numerical results reveal that the EE of the $\eta^*$-pairing state exhibits an approximately logarithmic scaling with subsystem size $l$, even in the band-touching limit ($t' = 2t$). Fig 3(a) shows the calculated EE as a function of subsystem size $l$ for various values of the intra-cell hopping $t'$. We find that the EE of the $\eta^*$-pairing states follows the logarithmic behavior, approaching the $\bar{\eta}$-pairing state (red line) as $\Delta$ increases, although deviations from the $\bar{\eta}$-pairing become apparent when the band gap decreases.

The modifications of the EE can be intuitively understood via the spatial structure of the pairing state. In the isolated flat band limit, the $\eta^*$-pairing is strictly confined to a single CLS:

$$|\eta_1^*\rangle \approx \frac{1}{2}\sum_i (c_{i,A,\uparrow}^\dagger - c_{i,B,\uparrow}^\dagger)(c_{i,A,\downarrow}^\dagger - c_{i,B,\downarrow}^\dagger)|0\rangle. \tag{28}$$

When the virtual tunneling process is accessed via interband coupling, the system develops a finite correlation length between the spin-up and spin-down electrons in the doublon. However, as the band gap $\Delta$ increases, the eigenstates are increasingly governed by the leading-order term $H_{\text{eff}}^{(1)}$, which dictates a strictly CLS confined doublon configuration. The contributions responsible for doublon broadening arise from higher-order Schrieffer-Wolff terms (specifically those beyond second order, see Appendix B.3) and are therefore suppressed by powers of $t/\Delta$ relative to the confinement energy. Consequently, the amplitude of spatially separated spin pairs vanishes asymptotically.

This effect is captured by the calculation of the two-point correlation function of $|\eta_1^*\rangle$, given by:

$$C_2(d) = \sum_{\substack{i,j \\ d(i,j)=d}} \sum_{\alpha,\beta} \left\langle c_{i\alpha\uparrow}^\dagger c_{j,\beta,\downarrow}^\dagger c_{i\alpha\uparrow} c_{j,\beta,\downarrow} \right\rangle, \tag{29}$$

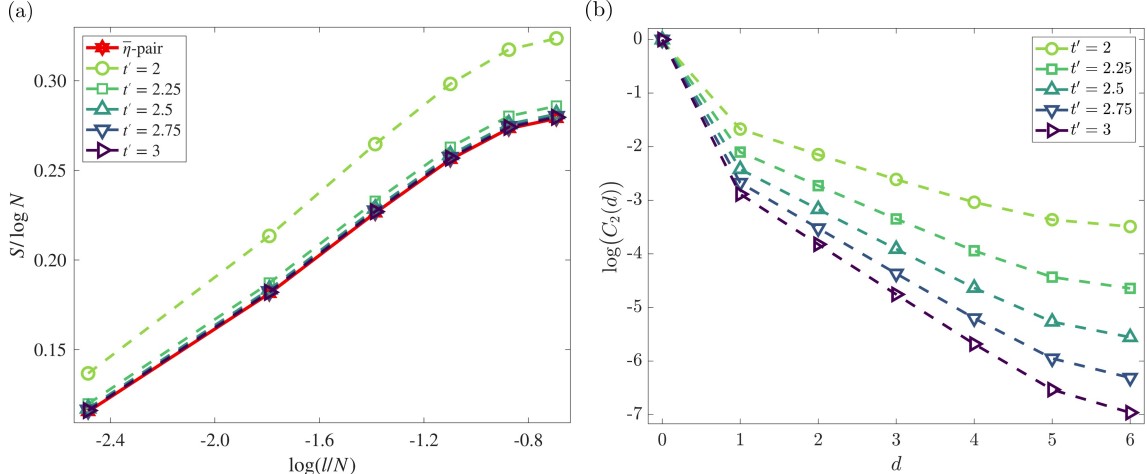

Figure 3: (a) EE scaling for the $\eta^*$-pairing states as a function of subsystem size $l$ (on a log scale) for a system of 12 unit cells and single $\eta^*$-pair ($|\eta_1^*\rangle$), with $t = 1$, $U = -1$, and various values of $t'$. The red solid line represents the EE of the $\bar{\eta}$-pairing state. Even in the band-touching limit ($t' = 2$), the $\eta^*$-pairing state displays a logarithmic EE scaling. (b) Logarithm of two-point correlation function $C_2(d)$ in the same system configuration with Fig. 3(a). Here, $d$ is the distance between the unit cells in PBC hosting the up and down spins of a doublon. The linear tail of the $\log(C_2(d))$ illustrates broadening of the doublon wave function and its exponentially decaying tail. Steepening of the $\log(C_2(d))$ at larger $t'$, indicates the localization of the doublon wave function, regaining spatial confinement of the $\bar{\eta}$-pairing states.

where the distance $d(i, j)$ between two unit cells $i$ and $j$ in PBC is defined by $d(i, j) = \min\{|i - j|, L - |i - j|\}$, with $L$ being the total number of unit cells in the system. Physically, $C_2(d)$ corresponds to the probability of observing the configuration of an up spin and a down spin separated by a distance $d$. Fig. 3(b) shows the calculated correlation function, which exhibits exponential decay as a function of the spatial separation of the doublon pair $d$. Importantly, in the large band gap limit, the two-point correlation function becomes sharp again, recovering the spatially confined doublon characteristic of the $\bar{\eta}$-pairing state. The exponential confinement of the doublon pair implies robust logarithmic EE behavior. However, when $t'$ approaches $2t + U$, the correlation function shows Friedel oscillations, signifying the breakdown of the logarithmic EE scaling.

## 5    Effective Repulsive Interactions

[Note: This section has been completely rewritten.]

In the limit of a perfectly isolated flat band ($\Delta \to \infty$), the single-doublon state $|\eta_1^*\rangle$ converges exactly to the ideal $\bar{\eta}$-pairing state. However, this convergence breaks down in the multi-pair sector. Ideally, the flat band hosts a manifold of degenerate doublon configurations, whose equal-weight superposition constitutes the $\bar{\eta}$-state. Our results (Fig. 5) indicate that the effective repulsive interaction arising from interband coupling acts as a perturbation that lifts this degeneracy. Consequently, even an infinitesimal repulsion selects spatially separated doublon configurations, preventing the formation of the uniform superposition characteristic of the ideal $\bar{\eta}$-pairing.

This effective interaction can be derived analytically via a fourth-order SW transformation. We find that the dominant contribution arises from the virtual hopping of doublons through

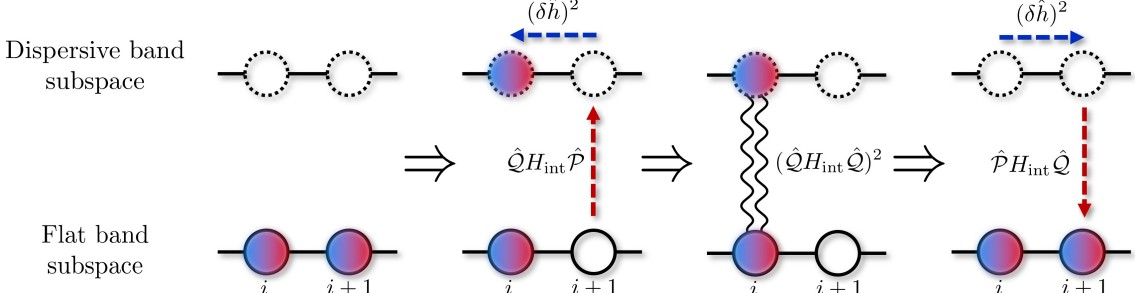

Figure 4: [Note: This figure is newly added.] Schematic representation of the virtual process responsible for the repulsive interaction between doublons. Solid circles represent the effective real-space sites of the flat band subspace, while dashed circles denote the corresponding sites in the dispersive subspace. The sequence depicts a flat band doublon at site $i$ undergoing a virtual excitation into the dispersive band (red upward arrow) and propagating to the neighboring site $i + 1$ (blue dashed arrow). At site $i + 1$, the virtual doublon interacts with the local flat band doublon via the Hubbard interaction (black wavy lines), before propagating back to site $i$ and relaxing (red downward arrow). This non-local sensing mechanism generates the effective nearest-neighbor repulsion.

the dispersive complementary band. The leading-order term in the effective Hamiltonian takes the form of a nearest-neighbor repulsion:

$$H_{\text{eff}}^{\text{int}} \approx \mathcal{C} \frac{t^4 U^4}{\Delta^7} \sum_{\langle i,j \rangle} n_i^D n_j^D, \tag{30}$$

where $n_i^D = \bar{n}_{i\uparrow} \bar{n}_{i\downarrow}$ is the doublon number operator and $\mathcal{C}$ is a positive constant. Physically, a doublon at site $i$ undergoes a virtual hopping to a neighboring site $j$. If the site $j$ is already occupied by the flat band doublon, the virtual state senses the flat band electrons via the Hubbard interaction, resulting in an energy increment proportional to $t^4 U^4$. This scaling ensures the interaction remains repulsive regardless of the sign of $t$ and $U$.

Crucially, this interaction is not strictly limited to the nearest neighbors. The resolvent operator $\hat{\mathcal{R}}$ generates a geometric series of hopping terms, $\sum_n (\delta \hat{h} / \Delta)^n$. Higher-order terms in this expansion allow virtual pairs to tunnel to distant sites before interacting and returning. Since each round trip for a doublon suppresses the amplitude by a factor roughly proportional to $t^4 / \Delta^4$, the effective interaction strength $V(d)$ exhibits an exponential decay with distance:

$$V(d) \sim V_0 \exp\left(-\frac{d}{\xi}\right), \quad \text{with } \xi \propto \left[\ln(\Delta/t)^4\right]^{-1}. \tag{31}$$

The repulsive interaction can be captured from the two-point density correlation function of doublons $|\eta_2^*\rangle$, given by:

$$C_4(d) = \sum_{\substack{i,j \\ d(i,j)=d}} \langle D_{i,j}^\dagger D_{i,j} \rangle \tag{32}$$

where $D_{i,j}^\dagger = \sum_{\alpha,\beta,\gamma,\delta} c_{i\alpha\uparrow}^\dagger c_{i,\beta,\downarrow}^\dagger c_{j,\gamma,\uparrow}^\dagger c_{j,\delta,\downarrow}^\dagger$ is the doublon pair creation operator at unit cells $i$ and $j$. Physically, $C_4(d)$ represents the probability of observing a configuration where doublons are separated by a distance $d$. Fig. 5(a) illustrates the behavior of the distance-dependent two-point density correlation function $C_4(d)$ for a system of 13 unit cells, with $t = 1$ and $U = -1$.

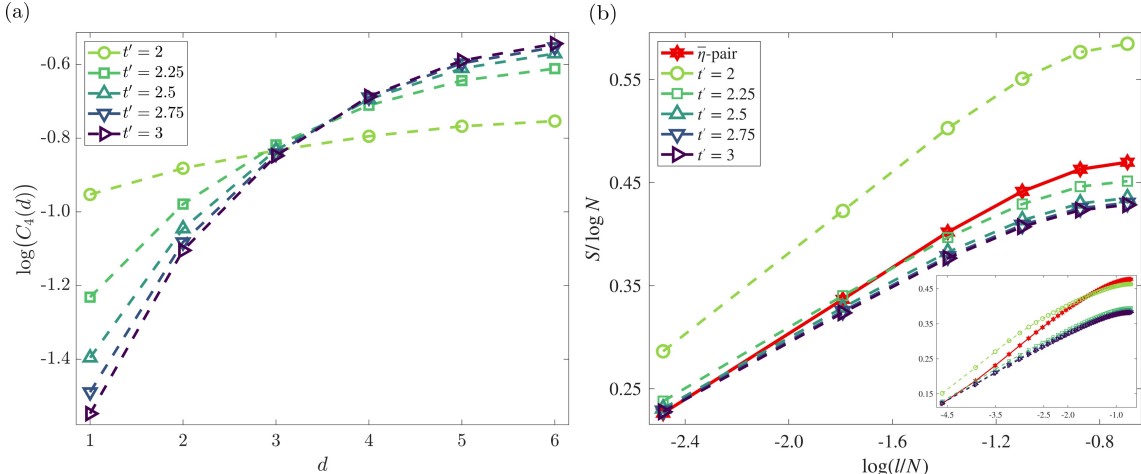

Figure 5: (a) Logarithm of four-point correlation function $C_4(d)$ in case of two $\eta^*$-pairs ($|\eta_2^*\rangle$) for a system of 13 unit cells, with $t = 1$, $U = -1$, and various values of $t'$. $d$ is the distance between unit cells occupied by localized doublons in PBC. It can be seen from the increase in $C_4(d)$ that the state exhibiting a distant doublon pair is more preferred due to the repulsive interaction between doublons. This tendency becomes more pronounced as the band gap ($t'$) becomes larger. (b) EE scaling for the $\eta^*$-pairing states in the case of three $\eta^*$-pairs ($|\eta_3^*\rangle$) as a function of subsystem size $l$ (on a logarithmic scale) for a system of 12 unit cells, with $t = 1$, $U = -1$, and various values of $t'$. The red solid line represents the EE of the exact multi $\bar{\eta}$-pairing state in the isolated flat band case. Although the $\eta^*$-pairing states converge in the large band gap limit, they do not coincide with the $\bar{\eta}$-pairing states. Inset shows finite-size scaling of the entanglement entropy for a state with 25 $\eta^*$-pairs in a system of $L = 100$ unit cells. These results were obtained using the Density Matrix Renormalization Group (DMRG) algorithm formulated within the Matrix Product State (MPS) framework, using the same Hamiltonian parameters. We can observe the same tendency holds even for larger systems size.

266 The result shows that $C_4(d)$ increases as doublons become more widely separated, reflecting
267 a preference for distant pair configurations. This behavior arises directly from the repulsive
268 interaction between $\eta^*$-pairs in Eq. (30).

269 Despite this deviation, key hallmarks of $\bar{\eta}$-pairing remain robust. In particular, the EE of
270 the multi-$\eta^*$ pairing states is even lower than that of $\bar{\eta}$-pairing states and follows a logarithmic
271 scaling with the subsystem size. Fig. 5(b) clearly shows that, even when the structure of equal
272 doublon superposition is lost, the EE retains a log-law behavior with an even lower value,
273 indicating that the underlying pairing order remains strong despite the breakdown of the exact
274 SU(2) symmetry.

275 This persistence of pairing order establishes the $\eta^*$-pairing state as the robust ground state
276 of the system in the attractive interaction regime. Specifically, when $t' > 0$, $U < 0$, and the
277 flat band is sufficiently isolated ($t' > 2|t| + U$), the effective Hamiltonian is dominated by
278 the first-order Schrieffer-Wolff term: $H_{\text{eff}} \approx H_{\text{eff}}^{(1)}$. Since $H_{\text{eff}}^{(1)}$ corresponds to the projected
279 attractive Hubbard interaction, the total energy is minimized by the configuration maximizing
280 the number of doublons. Since the $\bar{\eta}$-pairing state represents the manifold with the maximum
281 possible number of doublons, it corresponds to the lowest energy state in each number sector,
282 up to first order. Higher-order perturbative corrections, being suppressed by powers of the
283 band gap $\Delta$, introduce energy shifts that are insufficient to overcome the pairing energy gap
284 $|U|$.

285 Consequently, the preservation of these doublon pairs leads to an intriguing scenario where,
286 despite the effective inter-pair repulsion, the system's ground state transitions into a Bose–Einstein
287 condensate (BEC). The doublon pairs act as composite bosons, remaining robust against the
288 repulsive forces and condensing into a single macroscopic quantum state with off-diagonal
289 long-range order. Within a simple mean-field picture, the critical temperature $T_c$ is expected
290 to scale roughly with $|U|$, although a detailed prediction would require further analysis of the
291 band structure and interband coupling effects.

292 It is noteworthy that the effective electron mobility and repulsive interaction arises solely
293 from the dispersiveness of the complementary band (finite bandwidth $t$). In the previous study
294 on the $\eta$-pairing in the Creutz ladder with two flat bands [1], Cooper pair mobility is attributed
295 to the overlapping of the Wannier orbital characterized by quantum geometry of the flat band
296 Bloch functions. However, our result demonstrates that even in models where the flat band
297 geometry might be trivial, the kinetic dispersion of the dispersive band induces mobility of
298 electrons and a robust repulsion. This interaction forces the system into a density-ordered
299 configuration, preventing the $\eta^*$-states from converging to the $\bar{\eta}$-pairing even in the large gap
300 limit.

# 6 Discussion

302 In this work, we have investigated the deformation of flat band projection of $\eta$-pairing states
303 ($\bar{\eta}$-pairing states), due to the interband coupling, using the Creutz ladder as a case study.
304 Our analysis demonstrates that even when the flat band is not perfectly isolated, a subset of
305 eigenstates, the $\eta^*$-pairing states, retains key characteristics of $\bar{\eta}$-pairing. In the single doublon
306 regime, the $\eta^*$-pairing state converges to the exact $\bar{\eta}$-pairing state only in the large band gap
307 limit ($\Delta \to \infty$); for any finite gap, the SW transformation shows that virtual excitations into
308 the dispersive band produce an energy shift that deviates from the ideal value as the interaction
309 strength increases or the gap decreases. This quantifies the effect of interband coupling on the
310 pseudo-spin SU(2) symmetry.

311 We have characterized these $\eta^*$-pairing states through three key phenomena. First, the in-
312 terband coupling disrupts the exact spectrum generating algebra, resulting in modified energy
313 shifts driven by virtual excitations. Second, in real space, the two-point correlation function
314 shows that while the exact $\bar{\eta}$-pairing state features a sharply localized doublon confined to a
315 CLS, the $\eta^*$-pairing state exhibits a broadened doublon profile due to interband hybridization.
316 Third, we observe the emergence of an effective repulsive interaction between $\eta^*$-pairs. This
317 interaction, which scales as $\sim t^4 U^4 / \Delta^7$ and decays exponentially with distance, extends pre-
318 vious numerical findings of nearest-neighbor repulsion in the strong-coupling limit [39]. By
319 analytically deriving the exponential decay of the interaction strength $V(d)$, we explain the
320 anomaly in the multi-pair sector where $\eta^*$ eigenstates do not asymptotically converge to the
321 $\bar{\eta}$-pairing states even in the large band gap limit. Instead, the effective finite-range repulsion
322 selects spatially separated, density-ordered configurations over the equal weight superposi-
323 tions required for the SU(2) invariant state. Remarkably, despite these structural deforma-
324 tions, the entanglement entropy of the $\eta^*$-pairing state retains an approximately logarithmic
325 scaling with subsystem size $l$, distinguishing it from generic thermal eigenstates.

326 Furthermore, our results unveil the role of interband coupling in the dynamic proper-
327 ties of the flat band. While there are extensive studies on the flat band in the perturba-
328 tive regime [40, 41], recent literature has attributed flat band mobility to the quantum met-
329 ric [1, 2, 42–50] or disorder-induced inverse Anderson transitions [51–54], and characterized
330 regimes of interaction-induced localization [55–58], our work reveals a distinct mechanism.
331 We demonstrate that the effective hopping of electrons and doublons arises specifically from
332 the kinetic dispersion (finite bandwidth $t$) of the complementary band.

333 Collectively, our findings shed light on the interplay between interaction-driven pairing and

band structure effects in realistic flat band systems. The persistence of $\eta$-pairing signatures, such as low entanglement entropy and confined doublon correlations, even in the presence of interband coupling, suggests that strict band isolation is not an absolute requirement for robust pairing phenomena. This has significant implications for experimental platforms such as ultracold atoms, photonic lattices, and designer electronic systems, where perfect flat band isolation is often unattainable.

Optical lattices in cold-atom systems offer a controllable platform for realizing flat band structures. There have been experimental realizations of Lieb lattices in both bosonic and fermionic platforms by tuning the lattice unit cell [59, 60]. The control of the lattice distance can effectively tune the intra- and intercell hopping ratio $t'/t$ in addition to the interaction strength $U$. The wide range of controllability of the interaction strength ($U/t \approx 1 \sim 10$ via Feshbach resonance) can explicitly measure the dynamics of the many-body phase in a controllable manner [61–64]. In such setups, $\eta$-pairing states can be prepared via an adiabatic ramping process starting from a configuration with localized doublons [65]. Their nonthermal behavior can be signaled by a persistently high probability of spin–spin correlations, seen as a sharp peak in the pair momentum distribution [65, 66].

Future work may extend our analysis to the dynamical response of these pairing states and the exploration of similar phenomena in higher-dimensional flat band systems (such as Lieb and Kagome lattices). While the spatial dimensions of the system may result in qualitative changes in the dynamical behavior, it is important to note that the SW transformation does not alter the results. In our case, the Creutz lattice model exhibits a lower energy flat band with quadratic band touching. This energy band coincides with the Lieb lattice.

# 7 Acknowledgement

M.J.P. thanks Kyoung-Min Kim and Beom Hyun Kim for helpful discussions. This work was supported by the National Research Foundation of Korea (NRF) grant funded by the Korea government (MSIT) (Grants No. RS-2023-00218998). This work was supported by the BK21 FOUR (Fostering Outstanding Universities for Research) program through the National Research Foundation (NRF) funded by the Ministry of Education of Korea.

**Funding information**   M.J.P.: NRF RS-2023-00218998.

363    [Note: All Appendices are newly added.]

## A    Projected $\eta$-Pairing State

365    In the projected subspace, we define the $\bar{\eta}$-pairing state as:

$$
P_{\alpha\beta}(\mathbf{k}) = u_{\mathbf{k}} u_{\mathbf{k}}^{\dagger} = \begin{pmatrix} \frac{1}{2} & -\frac{1}{2} \\ -\frac{1}{2} & \frac{1}{2} \end{pmatrix} \tag{33}
$$

$$
\bar{c}_{\mathbf{R}_i \alpha} = \frac{1}{\sqrt{N}} \sum_{\mathbf{k}} e^{-i\mathbf{k}\cdot\mathbf{R}_i} \bar{c}_{\mathbf{k}\alpha}
$$

$$
= \frac{1}{\sqrt{N}} \sum_{\mathbf{k},\beta} e^{-i\mathbf{k}\cdot\mathbf{R}_i} P_{\alpha\beta}(\mathbf{k}) c_{\mathbf{k}\beta}
$$

$$
= \frac{1}{\sqrt{N}} \sum_{\mathbf{R}_j} \left( \frac{1}{\sqrt{N}} \sum_{\mathbf{k},\beta} e^{-i\mathbf{k}\cdot(\mathbf{R}_i - \mathbf{R}_j)} \right) P_{\alpha\beta} c_{\mathbf{R}_j \beta}
$$

$$
= \sum_{\beta} P_{\alpha\beta} c_{\mathbf{R}_i \beta}. \tag{34}
$$

366    Here, we used the $\mathbf{k}$ independence of $P_{\alpha\beta}(\mathbf{k})$. We henceforth denote $\bar{c}_{\mathbf{R}_i \alpha}$ as $\bar{c}_{i\alpha}$ for simplicity.
367    The complementary second quantization operators $\tilde{c}_{i\alpha}$ can be defined as $\tilde{c}_{i\alpha} = c_{i\alpha} - \bar{c}_{i\alpha}$. These
368    second quantization operators satisfy the following anticommutation relations:

$$
\{\bar{c}_{i\alpha\sigma}, \bar{c}_{j\beta\sigma'}^{\dagger}\} = P_{\alpha\beta} \delta_{ij} \delta\sigma\sigma'
$$

$$
\{\tilde{c}_{i\alpha\sigma}, \tilde{c}_{j\beta\sigma'}^{\dagger}\} = \delta_{\alpha\beta} \delta_{ij} \delta\sigma\sigma' - P_{\alpha\beta} \delta_{ij} \delta\sigma\sigma'
$$

$$
\{\bar{c}_{i\alpha\sigma}, \tilde{c}_{j\beta\sigma'}^{\dagger}\} = 0 \tag{35}
$$

369    Then we define the annihilation operator of the flat band subspace as:

$$
\bar{c}_{iA\sigma} = \frac{1}{2}(c_{iA\sigma} - c_{iB\sigma}) \tag{36}
$$

$$
\bar{c}_{iB\sigma} = -\frac{1}{2}(c_{iA\sigma} - c_{iB\sigma}), \tag{37}
$$

370    with the flat band $\bar{\eta}$-operator:

$$
\bar{\eta}^{\dagger} = \sum_{i\alpha} \bar{c}_{i\alpha\uparrow}^{\dagger} \bar{c}_{i\alpha\downarrow}^{\dagger}. \tag{38}
$$

## B    Schrieffer Wolff Transformation

### B.1    Resolvent Operator Formalism

373    We now consider the second-order Schrieffer Wolff transformation of the given Hamiltonian.
374    If we define the many-body projection operator to the flat band Hilbert-Fock space as $\hat{\mathcal{P}}$, then
375    the many-body projector to the complementary space is $\hat{\mathcal{Q}} = \mathbf{1} - \hat{\mathcal{P}}$. These projection operators
376    satisfy the following property:

$$
\hat{\mathcal{P}} \tilde{c}_{i\alpha\sigma}^{\dagger} = \tilde{c}_{i\alpha\sigma} \hat{\mathcal{P}} = 0, \tag{39}
$$

377 and $\hat{\mathcal{Q}}\mathcal{C}^\dagger$ ($\mathcal{C}\hat{\mathcal{Q}}$) is non-zero only when $\mathcal{C}$ ($\mathcal{C}^\dagger$) is a product of second quantization operators
378 containing the operator $\tilde{c}_{i\alpha\sigma}$ ($\tilde{c}_{i\alpha\sigma}^\dagger$).
379 We now compute the Schrieffer Wolff Hamiltonian up to second order:

$$H_{\text{eff}} = H_{\text{kin}}\hat{\mathcal{P}} + \hat{\mathcal{P}}H_{\text{int}}\hat{\mathcal{P}} + \frac{1}{2}\hat{\mathcal{P}}[\mathcal{L}(H_{\text{int}}), \mathcal{O}(H_{\text{int}})]\hat{\mathcal{P}}, \tag{40}$$

380 where the superoperators $\mathcal{O}(X)$ and $\mathcal{L}(X)$ acting on a generic operator X are defined as:

$$\mathcal{O}(X) = \hat{\mathcal{P}}X\hat{\mathcal{Q}} + \hat{\mathcal{Q}}X\hat{\mathcal{P}} \tag{41}$$

$$\mathcal{L}(X) = \sum_{i,j} \frac{|i\rangle \langle i| \mathcal{O}(X) |j\rangle \langle j|}{E_i - E_j} \tag{42}$$

381 The first-order Schrieffer Wolff Hamiltonian can be easily computed from Eq. 39 as follows:

$$
\begin{aligned}
\hat{\mathcal{P}}H_{\text{int}}\hat{\mathcal{P}} &= U\sum_{i\alpha} \hat{\mathcal{P}}c_{i\alpha\uparrow}^\dagger c_{i\alpha\downarrow}^\dagger c_{i\alpha\downarrow} c_{i\alpha\uparrow}\hat{\mathcal{P}} \\
&= U\sum_{i\alpha} \hat{\mathcal{P}}(\bar{c}_{i\alpha\uparrow}^\dagger + \tilde{c}_{i\alpha\uparrow}^\dagger)(\bar{c}_{i\alpha\downarrow}^\dagger + \tilde{c}_{i\alpha\downarrow}^\dagger)(\bar{c}_{i\alpha\downarrow} + \tilde{c}_{i\alpha\downarrow})(\bar{c}_{i\alpha\uparrow} + \tilde{c}_{i\alpha\uparrow})\hat{\mathcal{P}} \\
&= U\sum_{i\alpha} \bar{c}_{i\alpha\uparrow}^\dagger \bar{c}_{i\alpha\downarrow}^\dagger \bar{c}_{i\alpha\downarrow} \bar{c}_{i\alpha\uparrow}\hat{\mathcal{P}} = U\bar{H}_{\text{Hub}} \\
&= \bar{H}_{\text{int}}\hat{\mathcal{P}} \tag{43}
\end{aligned}
$$

For the second order, we begin by rewriting the second-order SW term in the resolvent operator form:

$$H_{\text{eff}}^{(2)} = \frac{1}{2}\hat{\mathcal{P}}[\mathcal{L}(H_{\text{int}}), \mathcal{O}(H_{\text{int}})]\hat{\mathcal{P}}$$

Let us consider the matrix element between two states $|p\rangle, |p'\rangle$ in the flat band (subspace $\mathcal{P}$):

$$\langle p|H_{\text{eff}}^{(2)}|p'\rangle = \frac{1}{2}\langle p| (\mathcal{L}(H_{\text{int}})\mathcal{O}(H_{\text{int}}) - \mathcal{O}(H_{\text{int}})\mathcal{L}(H_{\text{int}})) |p'\rangle$$

Since $|p\rangle$ and $|p'\rangle$ are in the $\mathcal{P}$ subspace, the intermediate states must be in the $\mathcal{Q}$ subspace (dispersive band) for the matrix element to be non-zero. Inserting the identity $\sum_q |q\rangle\langle q| = \hat{\mathcal{Q}}$:
1. First Term ($\mathcal{LO}$): The operator $\mathcal{O}(H_{\text{int}})$ acts on $|p'\rangle$ to create a state in Q. Then $\mathcal{L}$ acts on the transition operator $|p\rangle\langle q|$.

$$\frac{1}{2}\sum_q \frac{\langle p|H_{\text{int}}|q\rangle\langle q|H_{\text{int}}|p'\rangle}{E_p - E_q}$$

2. Second Term ($-\mathcal{OL}$): The operator $\mathcal{L}$ acts on $|q\rangle\langle p'|$ first.

$$-\frac{1}{2}\sum_q \langle p|H_{\text{int}}|q\rangle \left( \frac{\langle q|H_{\text{int}}|p'\rangle}{E_q - E_{p'}} \right) = \frac{1}{2}\sum_q \frac{\langle p|H_{\text{int}}|q\rangle\langle q|H_{\text{int}}|p'\rangle}{E_{p'} - E_q}$$

(Note the sign flip in the denominator from $E_q - E_{p'}$). Since $|p\rangle$ and $|p'\rangle$ are both in the flat band, they have the same energy $NE_{\text{flat}}$, with $N$ being the total number of electrons: $N = \sum_{i\alpha\sigma}(\tilde{n}_{i\alpha\sigma} + \bar{n}_{i\alpha\sigma})$. Therefore, the denominators of the two terms are identical:

$$\langle p|H_{\text{eff}}^{(2)}|p'\rangle = \sum_q \frac{\langle p|H_{\text{int}}|q\rangle\langle q|H_{\text{int}}|p'\rangle}{NE_{\text{flat}} - E_q}.$$

This sum over $q$ is exactly the definition of the resolvent operator restricted to the $\mathcal{Q}$ subspace:

$$H_{\text{eff}}^{(2)} = \hat{\mathcal{P}}H_{\text{int}}\hat{\mathcal{Q}}\hat{\mathcal{R}}\hat{\mathcal{Q}}H_{\text{int}}\hat{\mathcal{P}},$$

where $\hat{\mathcal{R}} = \hat{\mathcal{Q}} \frac{1}{NE_{\text{flat}} - H_{\text{kin}}} \hat{\mathcal{Q}}$ is the resolvent operator. To expand the denominator of the resolvent operator, we compute the effective kinetic Hamiltonian in the complementary subspace $\hat{\mathcal{Q}}$ as:

$$\hat{\mathcal{Q}} H_{\text{kin}} \hat{\mathcal{Q}} = t' \sum_{i\alpha\sigma} (\tilde{n}_{i\alpha\sigma} - \bar{n}_{i\alpha\sigma}) + t \sum_{\langle i,j \rangle\sigma} \sum_{\alpha,\beta} \left( \tilde{c}_{i\alpha\sigma}^{\dagger} \tilde{c}_{j\beta\sigma} + \text{H.c.} \right)$$
$$= t' \sum_{i\alpha\sigma} (\tilde{n}_{i\alpha\sigma} - \bar{n}_{i\alpha\sigma}) + \delta\hat{h}. \tag{44}$$

Using $\Delta = 2t'$:

$$t' \sum_{i\alpha\sigma} (\tilde{n}_{i\alpha\sigma} - \bar{n}_{i\alpha\sigma}) - NE_{\text{flat}} = 2t' \sum_{i\alpha\sigma} \tilde{n}_{i\alpha\sigma}$$
$$= \Delta\tilde{N} \equiv \tilde{\Delta}_{\tilde{N}}. \tag{45}$$

We can now perform a geometric series expansion of the resolvent operator as:

$$\frac{1}{NE_{\text{flat}} - H_{\text{kin}}} = \frac{1}{-\tilde{\Delta}_{\tilde{N}} - \delta\hat{h}} \approx -\frac{1}{\tilde{\Delta}_{\tilde{N}}} + \frac{1}{\tilde{\Delta}_{\tilde{N}}} \delta\hat{h} \frac{1}{\tilde{\Delta}_{\tilde{N}}}. \tag{46}$$

The leading-order term, corresponding to the zeroth order of the geometric series, gives the local renormalization of the on-site interaction:

$$H_{\text{local}}^{(2)} = -\frac{1}{\tilde{\Delta}_2} \hat{\mathcal{P}} H_{\text{int}} \hat{\mathcal{Q}} H_{\text{int}} \hat{\mathcal{P}}$$
$$= -\frac{U^2}{2\Delta} \sum_{i,\alpha} \bar{n}_{i\alpha\downarrow} \bar{n}_{i\alpha\uparrow}. \tag{47}$$

Here, we used

$$\hat{\mathcal{Q}} H_{\text{int}} \hat{\mathcal{P}} = U(\mathcal{S}_{\uparrow}^{\dagger} + \mathcal{S}_{\downarrow}^{\dagger} + \mathcal{D}^{\dagger}) \tag{48}$$

where $\mathcal{S}_{\uparrow}^{\dagger} = \sum_{i\alpha} \tilde{c}_{i\alpha\uparrow}^{\dagger} \bar{c}_{i\alpha\uparrow} \bar{n}_{i\alpha\downarrow} \hat{\mathcal{P}}$, $\mathcal{S}_{\downarrow}^{\dagger} = \sum_{i\alpha} \tilde{c}_{i\alpha\downarrow}^{\dagger} \bar{c}_{i\alpha\downarrow} \bar{n}_{i\alpha\uparrow} \hat{\mathcal{P}}$, and $\mathcal{D}^{\dagger} = \sum_{i\alpha} \tilde{c}_{i\alpha\uparrow}^{\dagger} \tilde{c}_{i\alpha\downarrow}^{\dagger} \bar{c}_{i\alpha\downarrow} \bar{c}_{i\alpha\uparrow} \hat{\mathcal{P}}$. Physically, $\mathcal{S}^{\dagger}$ annihilates a single flat band electron in the flat band doublon and creates a single virtual (dispersive band) electron, while $\mathcal{D}^{\dagger}$ annihilates a flat band doublon and creates a virtual doublon.

## B.2 Effective Electron Hopping

The first order of the geometric series gives the highest-order non-local term:

$$H_{\text{non-local}}^{(2)} = \hat{\mathcal{P}} H_{\text{int}} \hat{\mathcal{Q}} \left( \frac{\delta\hat{h}}{\tilde{\Delta}_{\tilde{N}}^2} \right) \hat{\mathcal{Q}} H_{\text{int}} \hat{\mathcal{P}}. \tag{49}$$

Using the interaction decomposition derived previously, $\hat{\mathcal{Q}} H_{\text{int}} \hat{\mathcal{P}} = U(\mathcal{S}_{\uparrow}^{\dagger} + \mathcal{S}_{\downarrow}^{\dagger} + \mathcal{D}^{\dagger})$, and noting that cross-terms involving $\mathcal{D}^{\dagger}$ vanish because $\delta\hat{h}$ cannot transition between single-particle and two-particle excited states in the $\hat{\mathcal{Q}}$ subspace without vanishing due to site exclusion, we focus on the $\mathcal{S}\delta\hat{h}\mathcal{S}^{\dagger}$ terms.

Consider the spin-up contribution $\mathcal{S}_{\uparrow} \delta\hat{h} \mathcal{S}_{\uparrow}^{\dagger}$:

$$\mathcal{S}_{\uparrow} \delta\hat{h} \mathcal{S}_{\uparrow}^{\dagger} = \frac{U^2 t}{\Delta^2} \sum_{l\lambda,m\mu} \sum_{\langle i,j \rangle} \sum_{\alpha\beta} \hat{\mathcal{P}} \bar{n}_{l\lambda\downarrow} \bar{c}_{l\lambda\uparrow}^{\dagger} \tilde{c}_{l\lambda\uparrow} \left( \tilde{c}_{i\alpha\uparrow}^{\dagger} \tilde{c}_{j\beta\uparrow} \right) \tilde{c}_{m\mu\uparrow}^{\dagger} \bar{c}_{m\mu\uparrow} \bar{n}_{m\mu\downarrow} \hat{\mathcal{P}}. \tag{50}$$

400  We evaluate the contraction of the dispersive operators using the correct anticommutation
401  relation $\{\tilde{c}_{x\eta\sigma}, \tilde{c}^\dagger_{yv\sigma'}\} = Q_{\eta v}\delta_{xy}\delta_{\sigma\sigma'}$, where $Q_{\eta v} = \delta_{\eta v} - P_{\eta v}$:

$$\langle 0|\tilde{c}_{l\lambda\uparrow}\tilde{c}^\dagger_{i\alpha\uparrow}\tilde{c}_{j\beta\uparrow}\tilde{c}^\dagger_{m\mu\uparrow}|0\rangle = \langle 0|\{\tilde{c}_{l\lambda\uparrow}, \tilde{c}^\dagger_{i\alpha\uparrow}\}\{\tilde{c}_{j\beta\uparrow}, \tilde{c}^\dagger_{m\mu\uparrow}\}|0\rangle$$
$$= (Q_{\lambda\alpha}\delta_{li})(Q_{\beta\mu}\delta_{jm}). \tag{51}$$

402  Substituting this back, the sum over $i, j$ collapses to $i = l, j = m$ (where $l, m$ must be nearest
403  neighbors). The sum over $\alpha, \beta$ involves the projector elements:

$$\sum_{\alpha\beta} Q_{\lambda\alpha}Q_{\beta\mu} = \left(\sum_\alpha Q_{\lambda\alpha}\right)\left(\sum_\beta Q_{\beta\mu}\right). \tag{52}$$

404  Using the specific form of the projector for the Creutz ladder, $Q = \frac{1}{2}\begin{pmatrix} 1 & 1 \\ 1 & 1 \end{pmatrix}$, we find the row
405  sum $\sum_\gamma Q_{\eta\gamma} = 1$ for any $\eta$. Thus, the sublattice factors sum to unity: $1 \cdot 1 = 1$.
406      The term simplifies to:

$$\mathcal{S}_\uparrow \delta\hat{h}\mathcal{S}^\dagger_\uparrow = \frac{U^2 t}{\Delta^2}\sum_{\langle l,m\rangle}\sum_{\lambda\mu}\bar{n}_{l\lambda\downarrow}\bar{n}_{m\mu\downarrow}\bar{c}^\dagger_{l\lambda\uparrow}\bar{c}_{m\mu\uparrow}. \tag{53}$$

407  Including the Hermitian conjugate from $\delta\hat{h}$ and the analogous spin-down term $\mathcal{S}_\downarrow \delta\hat{h}\mathcal{S}^\dagger_\downarrow$, the
408  total non-local correction is:

$$H^{(2)}_{\text{non-local}} = \frac{U^2 t}{\Delta^2}\sum_{\langle i,j\rangle}\sum_{\alpha\beta}\left(\bar{n}_{i\alpha\downarrow}\bar{n}_{j\beta\downarrow}\bar{c}^\dagger_{i\alpha\uparrow}\bar{c}_{j\beta\uparrow} + \bar{n}_{i\alpha\uparrow}\bar{n}_{j\beta\uparrow}\bar{c}^\dagger_{i\alpha\downarrow}\bar{c}_{j\beta\downarrow} + \text{H.c.}\right). \tag{54}$$

409  This effective Hamiltonian term represents a density-assisted hopping process: a particle with
410  spin $\sigma$ hops between neighboring sites $i$ and $j$ (amplitude $\propto tU^2/\Delta^2$), but only if both sites are
411  occupied by particles of the opposite spin $\bar{\sigma}$. This interaction introduces non-local correlations
412  that depend on the spatial separation of a pair and an electron.

### B.3  Effective Pair Hopping

414  We derive the effective kinetic energy governing the motion of $\eta$-pairs. We start from the
415  general form of the second-order expansion term in the resolvent, which involves the operator
416  $(\delta\hat{h})^2$.
417      We define the virtual pair creation operator $\mathcal{D}^\dagger_{i\alpha}$ on sublattice $\alpha$ of unit cell $i$:

$$\mathcal{D}^\dagger_{i\alpha} = \hat{\mathcal{Q}}\tilde{c}^\dagger_{i\alpha\uparrow}\tilde{c}^\dagger_{i\alpha\downarrow}\bar{c}_{i\alpha\downarrow}\bar{c}_{i\alpha\uparrow}\hat{\mathcal{P}}. \tag{55}$$

418  The effective Hamiltonian for the pair sector is given by:

$$H^{(2)}_{\text{pair}} = -U^2 \sum_{i,j}\sum_{\alpha,\beta}\mathcal{D}_{i\alpha}\left(\frac{(\delta\hat{h})^2}{\tilde{\Delta}^3_2}\right)\mathcal{D}^\dagger_{j\beta}. \tag{56}$$

419  To evaluate this, we first express the $(\delta\hat{h})^2$ in its most general form. Since $\delta\hat{h} = t\sum_{\langle x,y\rangle}$
420  $\sum_{\mu v\sigma}(\tilde{c}^\dagger_{x\mu\sigma}\tilde{c}_{yv\sigma} + \text{H.c.})$, the square term relevant for pair transport involves the product of the
421  up-spin and the down-spin hopping term:

$$(\delta\hat{h})^2\bigg|_{\text{pair}} = t^2\sum_{\langle x,y\rangle}\sum_{\langle u,v\rangle}\sum_{\mu v\rho\lambda}\left(\tilde{c}^\dagger_{x\mu\uparrow}\tilde{c}_{yv\uparrow}\tilde{c}^\dagger_{u\rho\downarrow}\tilde{c}_{v\lambda\downarrow} + (\uparrow\leftrightarrow\downarrow)\right). \tag{57}$$

422 Here, the indices $\langle x, y \rangle$ and $\langle u, v \rangle$ denote independent nearest-neighbor bonds. This operator
423 moves an up-spin from $y \to x$ and a down-spin from $v \to u$.

424 We now plug this operator into Eq. (56). The creation operator $\mathcal{D}_{j\beta}^\dagger$ prepares a virtual pair
425 at site $j$ ($\tilde{c}_{j\beta\uparrow}^\dagger \tilde{c}_{j\beta\downarrow}^\dagger |0\rangle$). The annihilation operator $\mathcal{D}_{i\alpha}$ projects the final state onto a flat band
426 doublon at site $i$, which requires annihilating a virtual pair at site $i$ ($\langle 0| \tilde{c}_{i\alpha\downarrow} \tilde{c}_{i\alpha\uparrow}$).

427 The matrix element involves the following term:

$$\langle 0| \tilde{c}_{i\alpha\downarrow} \tilde{c}_{i\alpha\uparrow} \left[ \tilde{c}_{x\mu\uparrow}^\dagger \tilde{c}_{y\nu\uparrow} \tilde{c}_{u\rho\downarrow}^\dagger \tilde{c}_{v\lambda\downarrow} \right] \tilde{c}_{j\beta\uparrow}^\dagger \tilde{c}_{j\beta\downarrow}^\dagger |0\rangle. \tag{58}$$

428 For the above term to be non-zero: 1. The annihilation operators in $(\delta\hat{h})^2$ must destroy the
429 particles created at $j$. Thus, the source indices must match $j$:

$$y = j \quad \text{and} \quad v = j. \tag{59}$$

430 2. The creation operators in $(\delta\hat{h})^2$ must create particles at $i$ to be detected by the projector.
431 Thus, the destination indices must match $i$:

$$x = i \quad \text{and} \quad u = i. \tag{60}$$

432 This proves that at order $t^2$, long-range pair hopping (e.g., to next-nearest neighbors) is for-
433 bidden; such processes would require four hopping operators ($\propto t^4$). Substituting this back
434 into the Hamiltonian and summing over the internal dispersive sublattice indices $\mu, \nu, \rho, \lambda$:

$$H_{\text{pair}}^{(2)} = -\frac{U^2 t^2}{\tilde{\Delta}_2^3} \sum_{\langle i,j \rangle} \sum_{\alpha\beta} \left[ \left( \sum_\mu Q_{\alpha\mu} \right) \left( \sum_\nu Q_{\nu\beta} \right) \left( \sum_\rho Q_{\alpha\rho} \right) \left( \sum_\lambda Q_{\lambda\beta} \right) \right] \bar{c}_{i\alpha\uparrow}^\dagger \bar{c}_{i\alpha\downarrow}^\dagger \bar{c}_{j\beta\downarrow} \bar{c}_{j\beta\uparrow} + \text{H.c.} \tag{61}$$

435 For the Creutz ladder, the projectors satisfy the identity $\sum_\mu Q_{\alpha\mu} = 1$. The term in the square
436 brackets simplifies to unity. Recalling the definition $\eta_i^\dagger = \sum_\alpha \bar{c}_{i\alpha\uparrow}^\dagger \bar{c}_{i\alpha\downarrow}^\dagger$, we recover the effective
437 nearest-neighbor pair-hopping Hamiltonian:

$$H_{\text{pair}}^{(2)} = -J_{\text{eff}} \sum_{\langle i,j \rangle} \eta_i^\dagger \eta_j, \quad \text{with } J_{\text{eff}} = \frac{U^2 t^2}{8\Delta^3}. \tag{62}$$

438 We note that the general expansion of $(\delta\hat{h})^2$ also includes terms where $x \neq u$ (spins hop
439 to different sites). However, these terms vanish in the second-order effective Hamiltonian
440 because the local Hubbard interaction in the projector $\mathcal{D}_{i\alpha}$ cannot simultaneously annihilate
441 spatially separated virtual particles ($i \neq k$). Thus, pair breaking is forbidden at this order.

## C   Higher order Schrieffer–Wolff Transformation

443 For simplicity, we focus on the terms describing strictly dispersive propagation, where all in-
444 termediate states lie in the complementary subspace $\mathcal{Q}$.

### C.1   Third-order Schrieffer–Wolff Transformation

446 The third-order effective Hamiltonian is derived from the commutator structure $H_{\text{eff}}^{(3)} =$
447 $\frac{1}{2}\hat{\mathcal{P}}[V_{\text{od}}, \mathcal{L}[V_{\text{d}}, \mathcal{L}(V_{\text{od}})]]\hat{\mathcal{P}}$, where $V_{\text{d}} = \hat{\mathcal{P}}V\hat{\mathcal{P}} + \hat{\mathcal{Q}}V\hat{\mathcal{Q}}$, $V_{\text{od}} = \hat{\mathcal{P}}V\hat{\mathcal{Q}} + \hat{\mathcal{Q}}V\hat{\mathcal{P}}$, and $V = H_{\text{int}}$. Explic-
448 itly expanding this in the eigenbasis reveals two distinct contributions. The first is the direct

propagation through the dispersive band, and the second is a renormalization correction due to the wavefunction overlap:

$$\langle p|H_{\text{eff}}^{(3)}|p'\rangle = \sum_{q,q'\in\mathcal{Q}} \frac{\langle p|V_{\text{od}}|q\rangle\langle q|V_{\text{d}}|q'\rangle\langle q'|V_{\text{od}}|p'\rangle}{(E_{\text{flat}}-E_q)(E_{\text{flat}}-E_{q'})}. \tag{63}$$

Expressed compactly using the resolvent operator $\hat{R}$, the final form is:

$$H_{\text{eff}}^{(3)} = \hat{\mathcal{P}}V\hat{R}V\hat{R}V\hat{\mathcal{P}}. \tag{64}$$

## C.2  Fourth-order Schrieffer–Wolff Transformation

For the fourth order, we focus on the direct scattering term which involves four interaction events. We start from the Schrieffer-Wolff expression $H_{\text{direct}}^{(4)} = -\frac{1}{2}\hat{\mathcal{P}}[V_{\text{od}},\mathcal{L}[V_{\text{d}},\mathcal{L}[V_{\text{d}},\mathcal{L}(V_{\text{od}})]]]\hat{\mathcal{P}}$. This generates a chain of three intermediate propagators. In the eigenbasis:

$$\langle p|H_{\text{eff}}^{(4)}|p'\rangle = -\sum_{q,q',q''\in\mathcal{Q}} \frac{\langle p|V_{\text{od}}|q\rangle\langle q|V_{\text{d}}|q'\rangle\langle q'|V_{\text{d}}|q''\rangle\langle q''|V_{\text{od}}|p'\rangle}{(E_{\text{flat}}-E_q)(E_{\text{flat}}-E_{q'})(E_{\text{flat}}-E_{q''})}. \tag{65}$$

Using the resolvent operator, it can be simplified as follows:

$$H_{\text{eff}}^{(4)} = -\hat{\mathcal{P}}V\hat{R}V\hat{R}V\hat{R}V\hat{\mathcal{P}}. \tag{66}$$

# D  Doublon-Doublon Interaction

We derive the specific interaction term where a virtual doublon created at site $i$ hops to site $j$, and both of its spin components interact with the corresponding flat band doublon at $j$ before returning. This requires expanding the effective Hamiltonian to include two insertions of the mixed interaction operator $V_{\text{mix}} = U\sum_\sigma \bar{n}_{j\sigma}\tilde{n}_{j\bar{\sigma}}$.

We consider the perturbation pathway:

$$H_{\text{eff}} \approx \hat{\mathcal{P}}H_{\text{int}}\hat{\mathcal{Q}}\left[\frac{1}{\tilde{\Delta}_{\tilde{N}}}(\delta\hat{h})^2\frac{1}{\tilde{\Delta}_{\tilde{N}}}V_{\text{mix}}\frac{1}{\tilde{\Delta}_{\tilde{N}}}V_{\text{mix}}\frac{1}{\tilde{\Delta}_{\tilde{N}}}(\delta\hat{h})^2\frac{1}{\tilde{\Delta}_{\tilde{N}}}\right]\hat{\mathcal{Q}}H_{\text{int}}\hat{\mathcal{P}}, \tag{67}$$

For $\hat{\mathcal{Q}}H_{\text{int}}\hat{\mathcal{P}}$, we only consider the operator $\mathcal{D}^\dagger$ at site $i$, $\mathcal{D}_{i\alpha}^\dagger$, which excites a flat band doublon into the dispersive band at site $i$.

Step 1: Virtual Pair Creation and Hopping ($i \rightarrow j$) The interaction $\mathcal{D}_{i\alpha}^\dagger$ creates a virtual pair at $i$, and the second-order hopping $(\delta\hat{h})^2$ moves both spins to site $j$:

$$|\psi_j\rangle \propto t^2\tilde{c}_{j\uparrow}^\dagger\tilde{c}_{j\downarrow}^\dagger|\text{flat}\rangle_i. \tag{68}$$

At this stage, the virtual state contains one dispersive electron of each spin at site $j$.

Step 2: Sequential Interaction ($V_{\text{mix}}$ twice) The mixed interaction $V_{\text{mix}}$ acts on this state. Since the virtual state has $\tilde{n}_{j\uparrow} = 1$ and $\tilde{n}_{j\downarrow} = 1$, we focus on the cross-term contribution from $V_{\text{mix}}\ldots V_{\text{mix}}$ where each $V$ picks up a different spin component.

First application of $V_{\text{mix}}$ (picking up spin-up interaction):

$$V_{\text{mix}}|\psi_j\rangle \rightarrow (U\bar{n}_{j\downarrow}\tilde{n}_{j\uparrow})|\psi_j\rangle = U\bar{n}_{j\downarrow}|\psi_j\rangle. \tag{69}$$

Second application of $V_{\text{mix}}$ (picking up spin-down interaction):

$$V_{\text{mix}}(U\bar{n}_{j\downarrow}|\psi_j\rangle) \rightarrow U^2\bar{n}_{j\downarrow}(\bar{n}_{j\uparrow}\tilde{n}_{j\downarrow})|\psi_j\rangle = U^2\bar{n}_{j\downarrow}\bar{n}_{j\uparrow}|\psi_j\rangle. \tag{70}$$

This sequence explicitly generates the operator product $\bar{n}_{j\uparrow}\bar{n}_{j\downarrow}$. The virtual pair interacts with both background spins at site $j$.

Step 3: Return and Annihilation ($j \to i$) The pair hops back to $i$ via $(\delta\hat{h})^2$ and is annihilated by $B_i^\dagger$. Combining the remaining flat band creation/annihilation operators at site $i$ gives $\bar{n}_{i\uparrow}\bar{n}_{i\downarrow}$.

The final effective Hamiltonian should then take the form of:

$$H_{\text{rep}} \propto \frac{t^4 U^4}{\Delta^7}(\bar{n}_{i\uparrow}\bar{n}_{i\downarrow})(\bar{n}_{j\uparrow}\bar{n}_{j\downarrow}). \tag{71}$$

In the notation of doublon number $n^D = n_\uparrow n_\downarrow$, this result is:

$$H_{\text{rep}} \approx \mathcal{C}\frac{t^4 U^4}{\Delta^7}\sum_{\langle i,j\rangle} n_i^D n_j^D, \tag{72}$$

where $\mathcal{C}$ is a positive constant. The resulting positive-definite term ($U^4$) explains the universal repulsion between $\eta$-pairs.

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
