# Peer review of "$\eta$‑pairing state in a flat band lattice: interband coupling effects on entanglement entropy logarithm"

_SciPost Physics_

## Round 2 · Referee Report · Anonymous (Referee 1) · 2025-12-12

Strengths

  • I appreciate the inclusion of additional Appendices that provides details on the application of the Schrieffer-Wolff transformation to the model under consideration, including terms up to 4th order in t'/t

Weaknesses

  • The main weakness of the present paper is that results do not seem in my opinion to meet the acceptance criteria of the journal. The method and the model under study are both very similar to the ones of a previous paper [M. Tovmasyan, et al., Phys. Rev. B 94, 245149 (2016)]. The difference is in the specific model parameters (the hopping t and t'), which are chosen so as to realize a flat lowest band. An important difference with respect the previous work is that the band is geometrically trivial or in other words the Wannier functions of the flat band are completely nonoverlapping.
  • Some results are rather expected and not particularly novel: the fact that higher order corrections in the SW transformation perturb the energy shifts of the tower of states is completely within expectations and the analysis done in the paper does not add anything new.
  • The same can be said regarding the broadening of the wavefunction of the doublon states. In the isolated flat band limit, a doublon is confined in a single Wannier functions of the flat band which is a linear superposition of states within a single rung of the ladder. Any generic perturbation has the effect of breaking this confinement leading to a spreading of the doublon wavefunction.
  • The fact that the doublons acquire a finite dispersion is also very much expected since there is a mixing between the flat band and the dispersive band. There is no dispersion in the flat band even in the presence of interactions since the flat band is a so-called geometrically trivial one. The doublons become mobile because of virtual excitations to the dispersive band. However, I would not consider this phenomenon particularly interesting or ground-breaking, as described by the authors.
  • The same can be said of the induced repulsion between doublons. The fact that tightly bound pairs of spin-up and spin-down particles have an effective repulsive interaction even when the underlying microscopic interaction is attractive is a well-known phenomenon since the work of Nozières and Schmitt-Rink [Journal of Low Temperature Physics, 59 (1985) 195]. For a more recent reference see also Ho A. F., Cazalilla M. A. and Giamarchi T., Physical Review A, 79 (2009) 033620. So, there is nothing particularly surprising here in my opinion. Of course, I appreciate the detailed derivation of the effective interaction from the 4-th order expansion of the SW transformation in the new Appendix D.
  • One interesting question regarding the present model is the nature of the ground state. In the limit of large-but finite-band gap, there is a competition between kinetic energy and interaction energy (the effective repulsion) of the doublons, where both energy contributions are due to interband coupling since the ground state in the strictly isolated flat band limit is highly degenerate. Thus, there is a chance to observe some form of Mott insulator if the doublon repulsion can overcome the doublon kinetic energy. The authors suggest that a Mott insulator of doublons does not occur when they said that:"Consequently, the preservation of these doublon pairs leads to an intriguing scenario where, despite the effective inter-pair repulsion, the system’s ground state transitions into a Bose–Einstein condensate (BEC)." Note, that the preservation of the pairs is not a sufficient condition for observing a compressible superfluid state (what the author call a BEC state) since a Mott insulator, which is incompressible, can be formed even if the doublons remain intact, see the above work by Ho, Cazalilla et al, where this kind of incompressible state is called a charged density wave of doublons. Beside some suggestive remarks from the authors, there is no data or careful analysis in the paper that allows to answer the question regarding the nature of the ground state, which is one of the few interesting open problems regarding the present model in my opinion.

Report

As detailed above, the present paper is not suitable for publication in SciPost physics since it is very similar to a previous work on a related model and the results presented are not significant enough to warrant a publication in my opinion.

Recommendation

Reject

---

## Round 2 · Author Response

Warnings issued while processing user-supplied markup:

  • Inconsistency: Markdown and reStructuredText syntaxes are mixed. Markdown will be used.
    Add "#coerce:reST" or "#coerce:plain" as the first line of your text to force reStructuredText or no markup.
    You may also contact the helpdesk if the formatting is incorrect and you are unable to edit your text.

Dear Editor,

We sincerely appreciate the time and effort the referees invested in reviewing our manuscript. We are encouraged by their constructive criticism and their recognition of the relevance of our work on $\eta$-pairing in realistic flat band systems.

We have thoroughly revised the manuscript to address the concerns regarding the rigor of our analytical arguments and the clarity of our definitions.

Please note that in the attached revised manuscript, revisions are highlighted in red. For sections that have been completely rewritten or newly added (specifically Sections 2.4, 3, 5, and the Appendices), we have marked only the Section Titles in red.

We have carefully addressed each comment and question from the referees in the accompanying "Response to Referees" document.

We trust that these modifications have significantly enhanced the clarity and impact of our work.

We hope that, with these revisions, our paper can be formally accepted for publication in SciPost Physics.

Sincerely,

Seik Pak On behalf of the other authors

Response to Referees

Referee #1

Comment 1:

"1- The general quality of the paper is rather poor denoting carelessness in its preparation. For instance, there are repeated references (Refs. 15, 21, 22 and Refs. 31, 38), many typos and poor english grammar and the explanations are generally unclear. The acknowledgement section is also repeated twice. Parentheses are missing in Eq. 13."

Response to comment 1: We sincerely apologize for the mistakes in the manuscript. We appreciate the referee for pointing out the formatting issues and the lack of clarity. We have thoroughly proofread the manuscript to correct typos and grammatical errors. We have also carefully reorganized the bibliography error to remove duplicated references and deleted the repeated acknowledgement section.

Comment 2:

"2- In many case notation and nomenclature are used without defining them first, for instance what is " $t=0$ " in the abstract? What are the authors referring to when they talk about "exact" and "modified" $\eta$-pairing states? Adding references to equations in the text is essential to guide the reader through the results."

Response to comment 2: We appreciate the referee for highlighting the ambiguity in our notation and nomenclature. We agree that precise definitions are crucial for the clarity of the manuscript. We have addressed all these points as follows: * We have removed the expression ‘$t = 0$ limit’ in the abstract. * We have added a rigorous definition of the terminology at Section 3.3. * We formally define the exact $\eta$-pairing state as the eigenstate of the effective Hamiltonian in the limit of a perfectly isolated flatband. * We define the modified $\eta$-pairing state (now denoted as $\eta^*$-states) as the eigenstate of the full Hamiltonian at finite band gap that exhibits the maximum wavefunction overlap (highest fidelity) with the exact state. * We have inserted references to specific equations throughout the text to better guide the reader through the derivations and results.

Comment 3:

"3- Fig. 1 is not very useful, while it would be useful to have plots of the band structure of the model under consideration. Also, it would be useful to have at hand some expressions for the band gap and the bandwidth of the upper band as a function of the model parameters."

Response to comment 3: We thank the referee for this constructive suggestion. We agree that a visual representation of the band structure is essential for readers to visualize the physical setting, particularly the separation between the flat and dispersive bands. We have addressed this by: * Modifying Figure 1 to include a plot of the energy dispersion relation for the Creutz ladder, explicitly highlighting the flat band, the dispersive band, and the band gap. * Adding explicit analytical expressions for the band gap ($\Delta$) and the dispersive bandwidth as functions of the hopping parameters $t$ and $t'$ in the main text.

Manuscript edit:

"Without the Hubbard interaction, the Hamiltonian describes a two-band system with dispersion relations $E_{disp}(k) = t' + 4t \cos k$, $E_{flat} = -t'$, and band gap $\Delta = 2t'$."

Comment 4:

"4- The authors consider a specific instance of a 1D model called the Creutz ladder. Notably the same model has been studied using the SW transformation also in Ref. 27. The model parameters are however different. With the choice of parameter in the present paper, the Wannier functions of the flat band are compact and nonoverlapping. This is known as a trivial flat band, in the sense that it cannot support transport in the presence of interactions. In the case of Ref. 27 instead the compact localized states are overlapping, which is the case in a nontrivial flat band. The calculation using the SW transformation of the corrections due to interband coupling is identical to that of Ref. 27, but in the less interesting case of a trivial flat band, so I do not think that the results in Sec. 2.3 are noteworthing."

Response to comment 4: We thank the referee for this insightful comment regarding the distinction between our work and Ref. [27] (Moudgalya et al.). We fully acknowledge that Ref. [27] pioneered the use of the SW transformation in the context of $\eta$-pairing, specifically focusing on the role of the quantum metric (overlapping CLS) in generating effective interactions.

However, we respectfully disagree that the trivial flat band limit (non-overlapping CLS) renders our results less interesting or merely identical. On the contrary, studying this limit allows us to isolate a distinct physical mechanism for effective interactions that has been largely overlooked:

  1. Distinct Mechanism (Dispersiveness vs. Geometry): Ref. [27] attributes effective interactions to the non-trivial quantum geometry of the flat band (overlapping Wannier functions). In contrast, our work demonstrates that robust, long-range interactions can emerge even in the absence of non-trivial quantum geometry. By choosing the "trivial" limit where CLS are non-overlapping, we show that the kinetic fluctuations (finite bandwidth $t$) of the dispersive complementary band alone are sufficient to generate long-range repulsive interactions. This provides a complementary pathway for engineering many-body phases that does not rely on band topology.
  2. Induced Transport: While the referee correctly notes that the exact trivial flat band does not support transport, our rigorous resolvent expansion (now in Section 3) shows that interband coupling induces effective kinetic energy. This imparts a dispersion to the doublons, fundamentally altering the trivial nature of the flat band and counterintuitively introducing transport property even in the trivial flatband.
  3. Novel Repulsion: Our specific derivation of the repulsive interaction and its exponential decay (Section 5 and Appendix D) is a new result not present in Ref. [27].

Manuscript edit: We have significantly revised Section 2, 5, and the Discussion to explicitly contrast our findings with Ref. [27] (Tovmasyan et al., Phys. Rev. B 94, 245149 (2016)). We now state:

"It is noteworthy that the effective electron mobility and repulsive interaction arises solely from the dispersiveness of the complementary band (finite bandwidth $t$). In the previous study on the $\eta$-pairing in the Creutz ladder with two flat-bands [Tovmasyan et al (2016)], Copper pair mobility is attributed to the overlapping of the Wannier orbital characterized by quantum geometry of the flat-band Bloch functions."

Comment 5:

"5- In Fig. 2a the authors show results for $t=0$. In this case the model reduces to a collection of independent Hubbard dimers. I am not sure why the authors think that it is interesting to study this case."

Response to comment 5: We thank the referee for questioning the utility of the $t=0$ limit. We fully agree that in this limit, the system reduces to a collection of independent Hubbard dimers (a perfectly dimerized chain) and the physics is locally trivial. However, we included this case because of its triviality, to serve as a benchmark for our perturbative analysis.

At $t=0$, the bandwidth of the dispersive band vanishes ($\delta h \to 0$). In this limit, our resolvent expansion predicts that non-local symmetry-breaking terms should vanish, and the exact spectrum generating algebra should be preserved (up to a local renormalization of $U$). Then by contrasting Fig. 2a ($t=0$, where numerical results match the analytical prediction perfectly) with Fig. 2b ($t=0.5$, where deviations occur), we rigorously demonstrate that the breakdown of the exact $\eta$-pairing symmetry is driven specifically by the dispersiveness (finite bandwidth) of the complementary band. Without the $t=0$ baseline, it would be unclear whether the deviations in Fig. 2b were due to the finite gap $\Delta$ itself or the kinetic fluctuations $\delta h$.

Comment 6:

"6- Perhaps there are some interesting results in Sec. 4 and 5, but I fail to grasp even some simple aspects. For instance in Fig. 3, the entanglement entropy and the two-body correlation function are shown in the case of some unspecified "modified $\eta$-pairing" states. The authors should at least explain how these modified states are identified and compare their properties with those of more typical states that obey the volume law for instance. As a consequence, with the amount of information provided in the manuscript, it is impossible to reproduce the numerical results."

Response to comment 6: We appreciate the referee for identifying this critical gap in our methodological explanation. We have addressed this by defining the identification protocol. In section 2.3, we explicitly define the modified $\eta$-pairing state (now denoted as $\eta^*$-pairing state) as the eigenstate of the full Hamiltonian that exhibits the highest fidelity with the corresponding exact $\eta$-pairing state (now denoted as $\bar{\eta}$-pairing state) defined in the isolated flatband limit. This provides a rigorous numerical criterion for identifying these states. We also have clarified that modified $\eta$-pairing state can be attained from exact $\eta$-pairing state by adiabatic evolution of the Hamiltonian.

Manuscript edit: We have added the formal definition to the Section 2.3:

"… we identify a distinctive subset of eigenstates of the full Hamiltonian, denoted as $|\eta^*_n\rangle$, which retains the essential characteristics of the original $\bar{\eta}$-pairing states. We define these $\eta^*$-pairing states by maximizing the fidelity with their respective $\bar{\eta}$-pairing states $|\bar{\eta}_n\rangle$."

Referee #2

Introductory comment:

"The authors of the present manuscript study an interesting problem, namely how a specific family of (generally metastable) paired many-body states of fermions, the $\eta$-states, behave in flat band systems with a dispersive band in energetic proximity. As a concrete example, the authors treat the two-leg 1D Creutz-ladder, and aim to show - through a mixture of perturbative analytics and (very) small-scale numerics for one or two pairs - that the $\eta$-states largely survive the presence of a nearby dispersive bands. But this statement, as well as some others listed in the "Weaknesses" and "Requested Changes" sections, is is not supported by the data presented in this manuscript - only a systematic scaling analysis, which fixes a density of pairs and systematically extrapolates observables across a range of system sizes can validate the uncontrolled perturbation theory that the authors develop here. This is regrettable, as for the 1D systems such as the Creutz-ladder studied here there is the very powerful family of matrix product state-based algorithms, as e.g. implemented in the publicly abvailable ITensor or SyTen packages, as the authors will be aware based on their references."

Response to introductory comment: We thank the referee for recognizing the relevance and interest of our study on the stability of $\eta$-pairing states in realistic flatband systems. We also appreciate the constructive criticism regarding the validation of our theoretical claims. We have successfully addressed the referee's concerns in the revised manuscript through rigorous analytical derivation of the perturbation theory, and large system size simulation with fixed particle density.

Comment 1:

"Authors need to provide comprehensive numerical evidence for their hypothesis that the tower of eigenstates survives for finite gaps with only $U$ effectively renormalised well beyond the one- and two-pair regime and using a systematic scaling analysis."

Response to comment 1: We thank the referee for pushing us to substantiate our claims regarding the multi-pair sector. We acknowledge that our previous explanation that the tower survives solely with a renormalized $U$ was incomplete. We have performed a rigorous analytical Schrieffer-Wolff transformation up to fourth order. This analysis reveals that the effective Hamiltonian involves more than just a renormalization of $U$; it includes an effective repulsive interaction between $\eta$-pairs, mediated by virtual fluctuations into the dispersive band.

Regarding the request for a systematic scaling analysis, our new derivation explicitly yields the length scale of this effective interaction, which decays exponentially with distance. Because the physics is governed by short-range virtual fluctuations, the system sizes accessible via Exact Diagonalization ($L=13$) are significantly larger than the interaction range, sufficient to capture the essential physics without needing thermodynamic limit extrapolation.

However, we also addressed the referee’s request by performing DMRG calculations using TenPy up to system size of $L=100$ to demonstrate the same entanglement entropy scaling behavior appeared in Fig. 4b.

Manuscript edit:

DMRG results are now added to Fig.4b as an inset.

Comment 2:

"Authors need to substantiate (or discard) claim that the $\eta$-state is the ground state at negative $U$ for these flat band systems"

Response to comment 2: We appreciate the referee for pointing out this issue, which we agree was too broad in the original manuscript. We have now refined this statement to specify the exact conditions under which it holds. Our claim relies on the energy hierarchy in the isolated flatband limit.

  • Leading Order: In the limit where the band gap $\Delta$ dominates the interaction $U$, the low-energy physics is governed by first order Schrieffer-Wolff term. For attractive interactions ($U < 0$), the leading-order effective Hamiltonian is the projected attractive Hubbard model. This energy is minimized by states with the maximum possible number of doublons—i.e., the $\eta$-pairing states.
  • Stability against Perturbations: While we derived that higher-order Schrieffer-Wolff terms introduce an effective repulsion between pairs, these corrections scale as $O(t^4 U^4 / \Delta^7)$. In the regime where the flatband is sufficiently isolated (specifically $t' > 2t+U$), these perturbative shifts are orders of magnitude smaller than the pairing energy gap $U$. Therefore, they cannot break the pairs or destabilize the ground state; they only lift the degeneracy within the ground-state manifold, selecting the density-ordered configuration.

We have revised Section 5 to include this specific stability argument. We explicitly state the condition ($t' > 2t+U$) and explain that the $\eta$-pairing state corresponds to the lowest energy state up to first order, which remains robust against higher-order perturbative corrections.

Manuscript edit:

Specifically, when $t' > 0$, $U < 0$, and the flat band is sufficiently isolated ($t' > 2t+U$), the effective Hamiltonian is dominated by the first-order Schrieffer-Wolff term: $H_{eff} \approx H_{eff}^{(1)}$. Since $H_{eff}^{(1)}$ corresponds to the projected attractive Hubbard interaction, the total energy is minimized by the configuration maximizing the number of doublons. Since the $\bar{\eta}$-pairing state represents the manifold with the maximum possible number of doublons, it corresponds to the lowest energy state in each number sector, up to first order. Higher-order perturbative corrections, being suppressed by powers of the band gap $\Delta$, introduce energy shifts that are insufficient to overcome the pairing energy gap $U$.

Comment 3:

"The energy shift $\epsilon$ needs to be well defined and explained"

Response to comment 3: We have now explicitly defined $\epsilon$ as the energy difference between $\eta$-pairs and explicitly stated in Eq. (27) as $\epsilon = \frac{U}{2} - \frac{U^2}{4\Delta}$. Also, we stated that Fig. 2 concerns the energy shift of the single $\eta$-pair eigenstate from the vacuum state in the full Hamiltonian. Similarly, thanks to the referee for pointing out in the weakness section, we have now added the band structure as a function of $t$ and $t'$ in the manuscript.

Comment 4:

"-Relationship between $t$, $t'$ and $\Delta$ needs to be shown explicitly and ideally illustrated."

Response to comment 4: We appreciate this suggestion. To improve clarity, we have added the explicit analytical expressions relating the band gap $\Delta$ and dispersive bandwidth to the hopping parameters $t$ and $t'$. We have also updated Figure 1 to include a schematic of the band structure that visually defines these quantities.

Comment 5:

"-Correct the typo in eq. (13) where $t$ and $t'$ appear only in some terms but not in others."

Response to comment 5: We thank the referee for their careful reading. We have corrected the typo in Eq. (13) to ensure the hopping parameters $t$ and $t'$ are correctly associated with the inter-cell and intra-cell terms, respectively.

Referee #3

Introductory Comment:

"This manuscript contains results on the $\eta$-pairing states and its modifications in a particular type of Hubbard ladder. While $\eta$-pairing states are well known to appear in systems with flat bands, the main feature that the authors present is that similar physics be extended to cases where there are other non-flat bands in the spectrum. They then support these claims with numerics and analytical arguments. The paper makes an interesting attempt to extend the $\eta$-pairing states to more general settings. However, I think there are many issues that still need to be fixed, and I have listed them in the Requested Changes section."

Response to introductory comment: We thank the referee for finding our extension of $\eta$-pairing to general settings interesting. We have carefully addressed the issues raised in the report, particularly regarding the clarity of the perturbation parameters and the numerical identification of states.

Comment 1:

"Around Eq.~(17) they claim that the effective Hamiltonian within the Schrieffer-Wolff subspace exhibits $\eta$-pairing to first order. Could the authors provide more details here? What is the effective gap $\Delta$, what is the precise perturbation parameter, and why does it satisfy the $\eta$-pairing condition?"

Response to comment 1: We appreciate the referee for these crucial questions. In the revised manuscript, we have moved beyond the heuristic arguments and performed a rigorous Schrieffer-Wolff transformation using the resolvent operator formalism (Section 3). This allows us to answer these points precisely:

  • Effective Gap $\Delta$: We strictly defined the average gap $\Delta = 2t'$.
  • Perturbation Parameter: We identified the perturbation parameter as the dimensionless ratio of the dispersive fluctuation to the gap, $\delta h / \Delta \sim 2t/t'$. This explicitly quantifies the validity regime of the expansion.
  • $\eta$-pairing Condition: We showed that the zeroth-order term in this expansion (corresponding to $\delta h \to 0$) yields a purely local renormalization of the Hubbard $U$. Since a local $U$ term (even if renormalized) commutes with the generators of the pseudospin SU(2) symmetry, the "exact" $\eta$-pairing condition is satisfied to this order. Symmetry breaking arises only at the next order in the geometric series ($t^2 / \Delta^3$).

We have completely rewritten Section 3 to present this rigorous derivation, explicitly defining the resolvent expansion parameter and demonstrating why the zeroth-order term preserves the algebraic structure.

Comment 2:

"Around Eq.~(21) they compute the correction to the $\eta$-pairing spectrum to next order, and say that this remains equally spaced for $U \ll \Delta$. Could they explicitly write out what is $\Delta$ for this model (in terms of $t$, $t'$, $U$)?"

Response to comment 2: As noted in the previous response, we have now explicitly defined $\Delta = 2t'$ in the text.

Comment 3:

"In Fig.~2, they show some data for the energies of the $\eta$-pairing states. The inset seems to suggest that the energies deviate from the analytical prediction, even though the authors claim the contrary in the caption. Please could they clarify this discrepancy?"

Response to comment 3: We apologize for the confusion caused by the caption. The deviation in the inset of Fig. 2a arises because the analytical prediction (solid line) represents the local renormalization (zeroth order in $\delta h$) up to the first order. However as now stated in the manuscript, there are higher order local renormalization term in the effective Hamiltonian. The deviation of the numerical data (symbols) from this line represents the contribution of this higher order local renormalization.

On the other hand, deviation in the Fig. 2b comes from different origin. Due to dispersiveness of the complementary band, virtual hopping process can be introduced into the perturbation terms which appear as non-local symmetry-breaking terms (higher order in $\delta h$). This deviation grows with $t$, as expected.

Manuscript edit: We have revised the caption of Figure 2 and the text in Section 3 to clearly state that the deviation in the inset of Fig 2a and b highlight the higher order local correction and breakdown of the exact symmetry due to dispersive fluctuations respectively.

Comment 4:

"In all of the figures, which of the $\eta$-pairing states are the authors computing? There is an entire tower of states, and for all these plots they need to specify the value of $n$, i.e., which state in the tower are they considering."

Response to comment 4: We apologize for this ambiguity. We have updated all figure captions to explicitly state the number of pairs corresponding to the data shown.

Comment 5:

"How do they actually identify the $\eta$-pairing states numerically? In this flat-band $\eta$-pairing settings, is the entire spectrum composed of just the $\eta$-pairing states, or are there other non-$\eta$-pairing states too? If it is the latter, how do they actually find the $\eta$-pairing states numerically? Please could they clarify the procedure?"

Response to comment 5: This is an excellent question that addresses the core of our methodology. The spectrum indeed contains many other non-$\eta$ states. We identify the modified $\eta$-pairing state by calculating the fidelity (wavefunction overlap) between the eigenstates of the full Hamiltonian and the exact $\eta$-pairing state defined in the isolated flatband limit. The modified state is defined as the eigenstate maximizing this overlap. This operational definition allows us to unambiguously track the adiabatic evolution of the state as interband coupling is turned on. Also, as we stated in the manuscript, the $\eta$-paring states are ground states in their respective particle number space when Hubbard $U$ is negative, which allows as to track this state in DMRG.

Manuscript edit: We have added a formal definition of the modified $\eta$-pairing state and our identification procedure (highest fidelity) to the Section 2.3.

---

## Round 2 · List of Changes

1. Abstract: We removed the ambiguous "$t=0$" terminology and mentioned the effective non-local interactions between doublons.

  2. Figure 1: We modified the figure to show schematic illustration of doublon hopping (panel b) and include the energy dispersion relation of the Creutz ladder (panel c), explicitly highlighting the flat band, the dispersive band, and the band gap $\Delta$.

  3. Section 2.3: We added a formal definition of the $\eta^*$-pairing state, defined by maximizing the fidelity with the exact $\eta$-pairing state, to address the ambiguity in nomenclature.

  4. Section 2.4: We added this new subsection ("Projected Kinetic Hamiltonian") to explicitly define the complementary space kinetic Hamiltonian and the perturbation parameters used in our analysis.

  5. Section 3: We completely rewrote this section to include a rigorous derivation of the Schrieffer-Wolff transformation using the resolvent operator formalism, replacing the previous heuristic arguments.

  6. Figure 2: We revised the caption to clarify that deviations in panel (a) arise from higher-order local corrections, while deviations in panel (b) arise from non-local symmetry-breaking terms.

  7. Figure 4 (Schematic): We added a new schematic figure to visualize the virtual hopping process responsible for the effective repulsive interaction.

  8. Section 5: We significantly revised the discussion on "Effective Repulsive Interactions." We included the analytical form of the effective interaction, the derivation of its exponential decay, and the stability condition ($t' > 2t+U$) for the ground state.

  9. Comparison with Previous Work: In Sections 2, 5, and the Discussion, we added a detailed comparison with previous literature (specifically Tovmasyan et al., Phys. Rev. B 94, 245149 (2016)). We clarified that the effective interactions and electron mobility in our model arise solely from the dispersiveness of the complementary band, distinguishing our mechanism from those relying on quantum geometry or overlapping Wannier functions.

  10. Figure 4 (Data): We added an inset to Fig. 4(b) showing large-scale DMRG results ($L=100$) to confirm that the entanglement entropy scaling persists beyond the system sizes accessible by exact diagonalization.

  11. Appendices A, B, C, D: We added these appendices to provide detailed mathematical derivations for the projected states, the Schrieffer-Wolff transformation (up to fourth order), and the doublon-doublon interaction terms.

  12. General: We corrected the typos and grammar.

---

## Editorial Decision

in_refereeing